# Observational perspective on SSWs and blocking from EP fluxes

Kamilya Yessimbet[1,2], Andrea K. Steiner[1,2], Florian Ladstädter[1], Albert Ossó[1]

[1]Wegener Center for Climate and Global Change, University of Graz, 8010 Graz, Austria

[2]FWF-DK Climate Change, University of Graz, 8010 Graz, Austria

*Correspondence to*: Kamilya Yessimbet (kamilya.yessimbet@uni-graz.at)

**Abstract.** In this study, we examine eight major boreal Sudden Stratospheric Warming (SSW) events between 2007 and 2019 to understand the vertical coupling between the troposphere and stratosphere, as well as the relationship between SSWs and blocking events using Global Navigation Satellite System (GNSS) radio occultation (RO) observations. Our study covers the main aspects of SSW events, including the vertical structure of planetary wave propagation, static stability, geometry of the polar vortex, and the

occurrence of blocking events. To analyze wave activity and atmospheric circulation, we compute the quasi-geostrophic Eliassen–Palm (EP) flux and geostrophic winds. The results show that the observations represent the primary dynamic features in agreement with theory and previous studies and provide a detailed view of their vertical structure. We observe a clear positive peak of upward EP flux in the stratosphere prior to all SSW events. In seven out of eight events, this peak is preceded by a clear peak in the troposphere. Within the observed timeframe, we identify two types of downward dynamic interactions and the emergence of blocking events. During the 2007

and 2008 "reflecting" events, we observe a displacement of the polar vortex, along with a downward propagation of wave activity from the stratosphere to the troposphere during vortex recovery, coinciding with the formation of blocking in the North Pacific region. Conversely, in the other six SSW "absorbing" events from 2009 to 2019, characterized by vortex split, we observe wave absorption and the subsequent formation of blocking in the Euro-Atlantic region. The analysis of the static stability demonstrates an enhancement of the polar tropopause inversion layer as the result of SSWs, which was stronger for the absorbing events. Overall, our study provides an

observational view of the synoptic and dynamic evolution of the major SSWs, their link to blocking, and the impact on the polar tropopause.

## 1 Introduction

In winter, dynamical coupling between the troposphere and stratosphere, in particular with the stratospheric polar vortex (SPV), is an important source of surface climate variability. The coupling is mediated by wave–mean-flow interactions and often occurs via the

downward progression of zonal mean anomalies following large SPV anomalies (Baldwin and Dunkerton, 2001). These downward anomalies can induce a change in the tropospheric circulation with patterns that resemble the annular modes (Baldwin and Dunkerton, 1999; 2001). In another view, the downward influence is mediated by the reflection of planetary wave activity from the stratosphere into the troposphere (Hines 1974; Geller and Alpert, 1980; Perlwitz and Harnik, 2004; Matthias and Kretschmer, 2020; Messori et al. 2022).

In this study, we focus on connections between Sudden Stratospheric warming (SSW) events, i.e., extreme cases of SPV variability, and atmospheric blocking, i.e., persistent high-pressure systems interrupting the regular westerly flow at midlatitudes. We adopt two distinctive classifications for SSWs. Depending on the SPV geometry, the first classification categorizes them into displacement and split events. In the displacement type of SSWs, the polar vortex is displaced away from the pole, and in the split type, the polar vortex

breaks up into daughter vortices (Butler et al., 2017). The second classification divides SSWs into reflecting and absorbing events based on the planetary wave activity evolution (Kodera et al., 2016). Reflecting SSW events are characterized by the downward reflection of planetary waves from the stratosphere into the troposphere, whereas absorbing events indicate non-reflecting stratospheric conditions,

implying wave absorption by the stratosphere, during SSW recovery. While the former classification describes the SPV behaviour during the mature phase of SSWs, the latter describes the behaviour of planetary wave activity during the recovery phase of SSWs, with a more pronounced focus on their impact on the troposphere.

In terms of the impacts on the troposphere, reflecting SSW events have been linked to the occurrence of North Pacific blocking, while absorbing SSWs have been associated with a hemispheric near-surface pattern of relatively high pressure over the polar region and low pressure in the mid-latitudes that resembles the negative phase of the Arctic Oscillation (AO) (Kodera et al., 2016). Some studies have suggested that the split and displacement types of SSW events may also lead to different tropospheric responses (Mitchell et al., 2013). However, other studies, such as Maycock and Hitchcock (2015), have not found significant differences in the tropospheric impacts of the split and displacement events.

The atmospheric layer where the main dynamic coupling occurs is the upper troposphere and lower stratosphere (UTLS). An accurate representation of the vertical structure of the UTLS is known to be important for the resolution of atmospheric dynamics and circulation in coupled climate models (Gerber and Manzini, 2016). This, in part, underpins the rationale for employing Global Navigation Satellite System (GNSS) Radio Occultation (RO) data in our study, which are known for their stability, accuracy, and high vertical resolution within the UTLS (Steiner et al., 2020). There have been previous studies resolving important aspects of atmospheric dynamics from RO data, such as Leroy et al. (2007) calculating quasi-geostrophic Eliassen–Palm (EP) flux, Scherllin-Pirscher et al. (2014) and Verkhoglyadova et al. (2014) calculating geostrophic winds, Healy et al. (2020) retrieving quasi-biennial oscillation (QBO) zonal winds, and Kedzierski et al. (2020) studying Rossby waves.

In this study, we utilize globally distributed direct measurements of geopotential height and temperature fields from RO data. These measurements are used to compute geostrophic winds, blocking index, quasi-geostrophic EP flux, and static stability. The computed parameters are then used as a basis for our synoptic and dynamic analysis of SSW events that occurred between 2007 and 2019. Our main objective is to characterize the dynamics induced by these SSW events and examine their links to blocking events from a observational perspective. We focus on the analysis of the vertical aspects of the EP flux, as it plays a critical role in understanding how the stratospheric circulation responds to the upward propagation of planetary wave activity from the troposphere (Yessimbet et al., 2022a). Due to its high vertical resolution, RO is shown in this study to be particularly suitable for providing information for the dynamic and synoptic analysis of SSW events and blocking events.

We describe the RO data set and detail the employed methods in Sect. 2 and 3, respectively. Section 4 presents the results of all investigated SSW events from 2006 to 2019 with a detailed analysis of two representative SSW events in February 2008 and January 2019. Finally, we discuss and conclude our findings in Sect. 5.

**2 Data**

This study employs measurements from GNSS RO collected by various satellite missions, including CHAMP (Wickert et al., 2001), SAC-C (Hajj et al., 2004), GRACE (Beyerle et al., 2005; Wickert et al., 2005), MetOp (Luntama et al., 2008), and Formosat-3/COSMIC (Anthes et al., 2011). The GNSS RO method is based on the detection of radio signals transmitted by GNSS satellites, which are refracted by the Earth's atmosphere as they propagate through it to Low Earth Orbit (LEO) satellites. The measured signal phase changes are converted to bending angle profiles, and further to refractivity by an Abel transform. At high altitudes, the Abel integral requires

initialization with background data. Thermodynamic parameters are then computed under the assumption of a dry atmosphere ("dry" parameters). In moist air conditions (lower to middle troposphere, specifically in the tropics), the retrieval of (physical) temperature or humidity requires prior knowledge of the state of the atmosphere (e.g., Kursinski et al. 1995; 1996). Due to the involved background data, the retrieved RO temperature data exhibit larger uncertainties in lowermost moist parts of the troposphere and at high altitudes
(above about 30 km). For an overview of the retrieval process and the involved structural uncertainties see, e.g., Steiner et al. (2020). The RO measurements are of high quality with minimal structural uncertainty within the UTLS region, as highlighted by Scherllin-Pirscher et al. (2017) and Steiner et al. (2020).

In this work, we use geopotential height and physical temperature as a function of pressure, processed by the Wegener Center for Climate
and Global Change (WEGC) with the Occultation Processing System (OPS) version 5.6 (Angerer et al., 2017; Steiner et al., 2020) with the phase delay data derived at the University Corporation for Atmospheric Research/COSMIC Data Analysis and Archive Centre (UCAR/CDAAC).

Geostrophic wind fields can be derived from RO geopotential height fields (Scherllin-Pirscher et al., 2014; 2017). RO geostrophic wind
and gradient wind fields were found to capture all main wind features in our study. Compared to atmospheric analyses, wind differences are generally small (2 m/s) except near the subtropical jet (up to 10 m/s). There, RO winds underestimate actual winds due to the geostrophic and gradient wind approximations while RO retrieval errors have negligible effects (Scherllin-Pirscher et al., 2014).

The number of daily RO profiles from different missions varied over the period from 2006 to 2019, with the highest number of profiles
from 2007 to 2010 (> 2500 profiles per day) and a decrease in the number of profiles (from more than 2500 to less than 2000 profiles) from 2012 onwards (Figure S1) due to the exceeding of the lifetime of some of the RO missions (Fig. 5, Angerer et al., 2017).

Utilizing data from the available records spanning 2006 to 2019, we focus on the daily wintertime period from November to March. The vertical grid consists of 147 pressure levels from 1000 to 10 hPa (equidistant in altitude space at 200 m up to 20 km, and 500 m
above that). To generate $2.5° \times 2.5°$ gridded bins from the profile data, we employ a spatial and temporal weighting methodology. This involves applying Gaussian weighting according to the latitude-longitude distances of each profile in relation to the bin center, taking all profiles within 600 km from the center into account. An additional temporal Gaussian weighting of +/- 2 days around the given day is also applied. With this, we reduce the number of missing grid points while maintaining as much measurement information as possible. Thus, in the range of vertical pressure levels from 10 to 850 hPa, there are fewer than 10 missing grid points in the daily gridded fields,
with the number increasing towards the surface. For any remaining missing grid points, we use bilinear interpolation to fill in these gaps.

**3 Method**

This study applies a geostrophic approximation to derive winds directly from geopotential heights as it balances between pressure gradient and Coriolis forces. RO measurements allow retrieval of geopotential height as an independent vertical coordinate. The computation of geostrophic winds is based on Scherllin-Pirscher et al. (2014).

To study wave activity, we calculate the quasi-geostrophic EP flux based on Edmon et al. (1980). According to the standard definition of the EP flux, the meridional $F_\phi$ and vertical $F_p$ components are defined as

$$\{F_\phi, F_p\} = \{-a\, cos\varphi\, \overline{u'v'}, f\, a\, cos\varphi\, \frac{\overline{v'\theta'}}{\theta_p}\}, \tag{1}$$

where $a$ denotes the equatorial radius of the Earth, $p$ denotes log-pressure, $\varphi$ is the latitude, and $\theta$ is potential temperature. The Coriolis parameter $f$ equals $2\Omega sin\varphi$, where $\Omega$ is the Earth's angular velocity, $u$ and $v$ are the geostrophic meridional and zonal winds, the overlines denote zonal averages, and primes are deviations from zonal averages. Divergence of the EP flux equal to $\partial\phi F_\phi + \partial p F_p$ shows an acceleration of the zonal flow. The climatological EP flux for winter 2006 to 2019 is shown in Fig.1, where the convergence zone (blue shading) with upward-directed EP flux vectors is observed in the upper troposphere, which means the wave activity is intensified in the wintertime upper troposphere. For display purposes, EP flux vectors are scaled as follows:

$$\{F_\phi, F_p\} = cos\varphi\, \{\frac{F_\phi}{a\,\pi}, \frac{F_p}{10^5}\}, \tag{2}$$

according to commonly used methods (University of Reading, 2023; NOAA's Physical Sciences Laboratory, 2023).

The 3D wave activity flux, Plumb flux, presented in the Appendix B, is computed according to Plumb (1987; eq. 5.7). In addition, we also compute the eddy meridional heat flux $\overline{v'T'}$ at 100 hPa and averaged over 45-75° N similar to Kodera et al. (2016), who used it as a proxy for the vertical wave propagation between the troposphere and the stratosphere to characterize reflecting/absorbing SSW events.

The computation of anomalies is performed by subtracting the 2007-2019 daily mean climatology. A standard algorithm, based on 500 hPa geopotential height gradients, described in Brunner and Steiner (2017) and Brunner (2018), is used to calculate the blocking index. Two distinct blocking regions have been defined based on the highest frequencies of blocking during the wintertime: North Pacific (160° E to 160° W) and Euro-Atlantic (30° W to 45° E).

To examine further effects of the different SSW types, we compute static stability in form of the Brunt Väisälä frequency, $N^2$, defined as follows:

$$N^2 = -(pg^2 \frac{\partial\theta}{\partial p})/(RT\theta), \tag{3}$$

where g denotes the gravitational acceleration ($g = 9.81\ m\ s^{-2}$), R is the gas constant of dry air ($R = 287\ J\ kg^{-1}K^{-1}$), and T is temperature.

In our analysis, we also made comparisons of key parameters between RO and reanalyses (e.g., ERA5), such as the zonally averaged parameters, zonal-mean zonal wind and $\overline{v'T'}$ (Figure S2 and S3), which confirmed the consistency and the reliability of the RO-based dynamics.

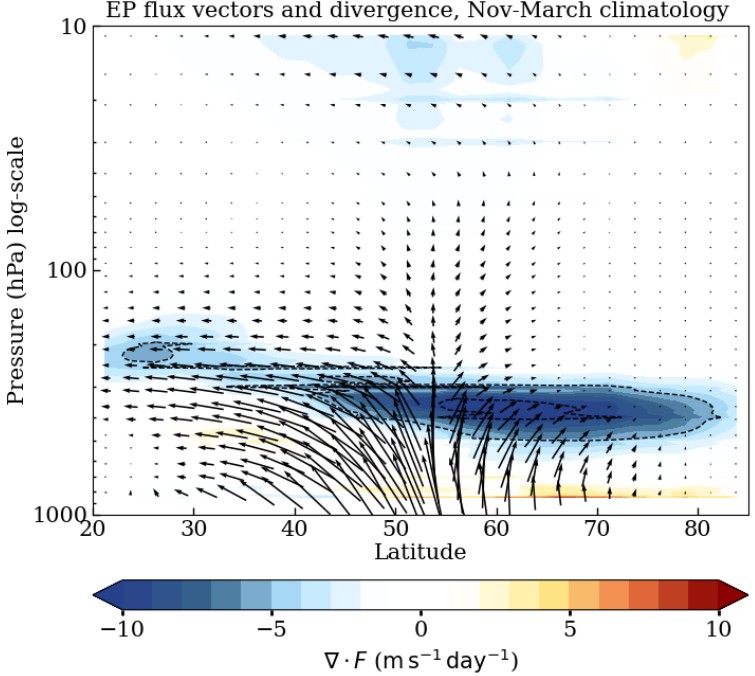

140

**Figure 1:** Climatological Eliassen–Palm flux vectors and divergence (shading) for November to March from 2006 to 2019. The vectors are plotted for every 5$^{th}$ pressure level.

For the definition of major SSW events, we adopt a commonly used definition according by Butler et al. (2017), which defines the central date of an SSW event as the day when the zonal-mean zonal wind at 10 hPa and 60° N, $\overline{u}$, changes from westerly to easterly. The diagnosed central SSWs are compared with the list of major midwinter SSWs in reanalysis products in the SSW Compendium dataset (NOAA CSL, 2024).

## 4 Results

Section 4.1 describes the commonalities in the key dynamics of the 2007 and 2008 winters. To illustrate this, we show the analysis results of the 2008 winter as a representative case study. Similarly, Sect. 4.2 presents the 2019 SSW event as an example to describe the commonalities between the 2009, 2010, 2013, 2016 and 2018 events. In Section 4.3, we analyze the vertical wave activity and static stability for all SSW events from 2007 to 2019.

### 4.1 Type I: SSW events in 2007 and 2008

One of the main commonalities of the 2008 and 2007 events was the SPV displacement. Figure 2a shows the vortex evolution during the main phases of the winter of 2008. During December and January, the vortex remained strong and symmetrically centred at the North Pole. In February, the vortex began to weaken, losing its symmetry and finally shifting from the pole towards Eurasia. The vortex weakening was marked by warming, with stratospheric polar cap (65-90°N) temperature anomalies exceeding the climatological mean and showing short-term fluctuation (Figs. 3a, b). While the vortex was displaced, positive polar cap temperature anomalies propagated

downwards to around 70 hPa. On 22 February, $\overline{u}$ turned easterly (Fig. 3c), marking the central date of a major SSW event. The reversal of $\overline{u}$ lasted for six days, followed by a period of recovery and gradual cooling. A similar pattern was observed in the winter of 2007, with a vortex displacement and $\overline{u}$ reversal on 24 February that lasted only four days and was also accompanied by fluctuating stratospheric polar cap temperature anomalies (Figs. A1a, A2a, b).

Another common feature of the 2008 and 2007 SSW events was a relatively short (about 12 and 13 days) pulse of wave activity propagation responsible for the $\overline{u}$ reversal, and its significant decrease (marked by a negative $\overline{v'T'}$) during the first week of the SSW recovery phase. The onset of the SSW recovery phase is defined as the date when the maximum North Pole (80-90°N) temperature anomalies at 50 hPa are reached (Kodera et al., 2016).

For the 2008 event, $\overline{v'T'}$ at 100 hPa and averaged across 45-75° N is shown in Fig. 3c. The behaviour of the $\overline{v'T'}$ is coherent with $\overline{u}$: as $\overline{v'T'}$ increases, indicating upward propagation of planetary wave activity, the wind speed decreases. From the beginning of February, a weakening of $\overline{u}$ occurred concurrent with the occurrence of two consecutive $\overline{v'T'}$ peaks with a two-day lag between them. The two-day lag between these two pulses may be an indication that the upward wave activity was suppressed and then resumed. The reversal of $\overline{u}$ occurred during the second $\overline{v'T'}$ peak, which lasted 12 days from 15 to 26 February. In the first week of the SSW recovery phase, the $\overline{v'T'}$ was characterized by negative values.

The time-height representation of the vertical component of EP flux as a function of pressure $F_p$ is shown in Fig. 3d. Since 7 February, there was a negative $F_p$ anomaly below 300 hPa, indicating an intensification of wave activity. Following this, a mildly negative $F_p$ pattern was observed, first between 300 and 100 hPa and then above 20 hPa with a delay of 2 days. On 15 February, another (less intense) negative $F_p$ anomaly appeared below 400 hPa, later extending to about 100 hPa, which indicates the upward propagation of wave activity. Between 18 February and the central date of the SSW event, a pronounced negative $F_p$ anomaly was noticeable above 100 hPa. Following the central SSW date, a notably substantial positive $F_p$ anomaly became apparent below 300 hPa, succeeded by a minor positive $F_p$ anomaly within the stratosphere. Starting from 29 February, the second peak of the positive $F_p$ anomaly maximized below the 300 hPa level and extended throughout the entirety of the atmospheric column. These instances of positive $F_p$ anomaly peaks were consistent with the negative $\overline{v'T'}$ peak observed from 27 February to 6 March, as illustrated in Fig. 3c. The negative $\overline{v'T'}$ peak suggests a substantial reduction in the propagation of planetary wave activity into the stratosphere or a downward propagation of wave activity due to the reflection of these waves from the stratosphere into the troposphere.

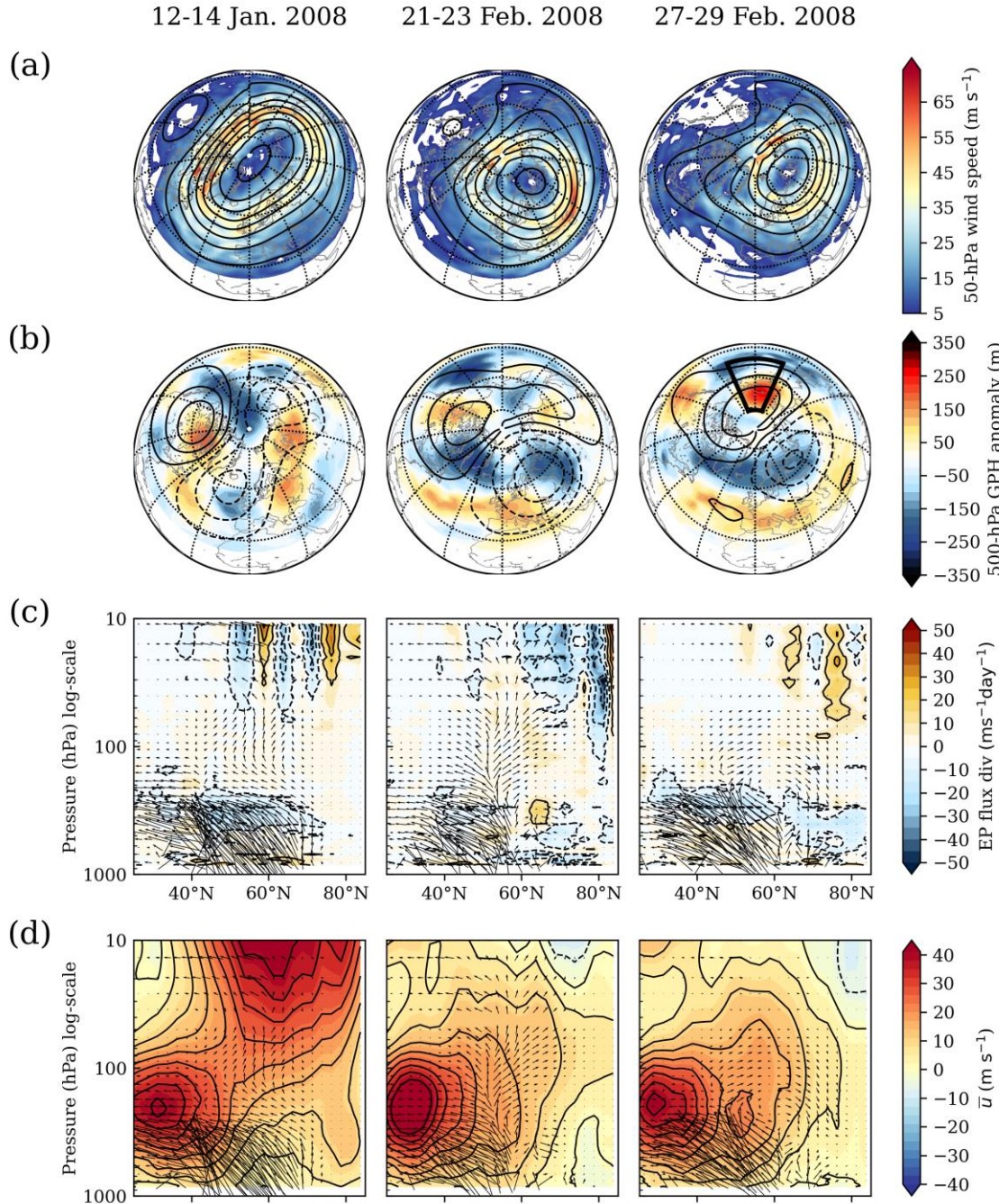

**Figure 2:** Evolution of the SSW 2008 shown for the three dates: before the SPV displacement (left), during its displacement (center) and during its recovery (right). (a) 50 hPa wind speed (shading) and 50 hPa geopotential height (contours). (b) 500 hPa geopotential height anomaly (shading) and 50 hPa geopotential height anomaly (contours). The black box indicates the North Pacific blocking region selected. (c) Meridional cross-sections of Eliassen–Palm flux vectors and divergence (shading). (d) Meridional cross-sections of Eliassen–Palm flux vectors and zonal wind (shading). The vectors are plotted for every 5$^{th}$ pressure level.

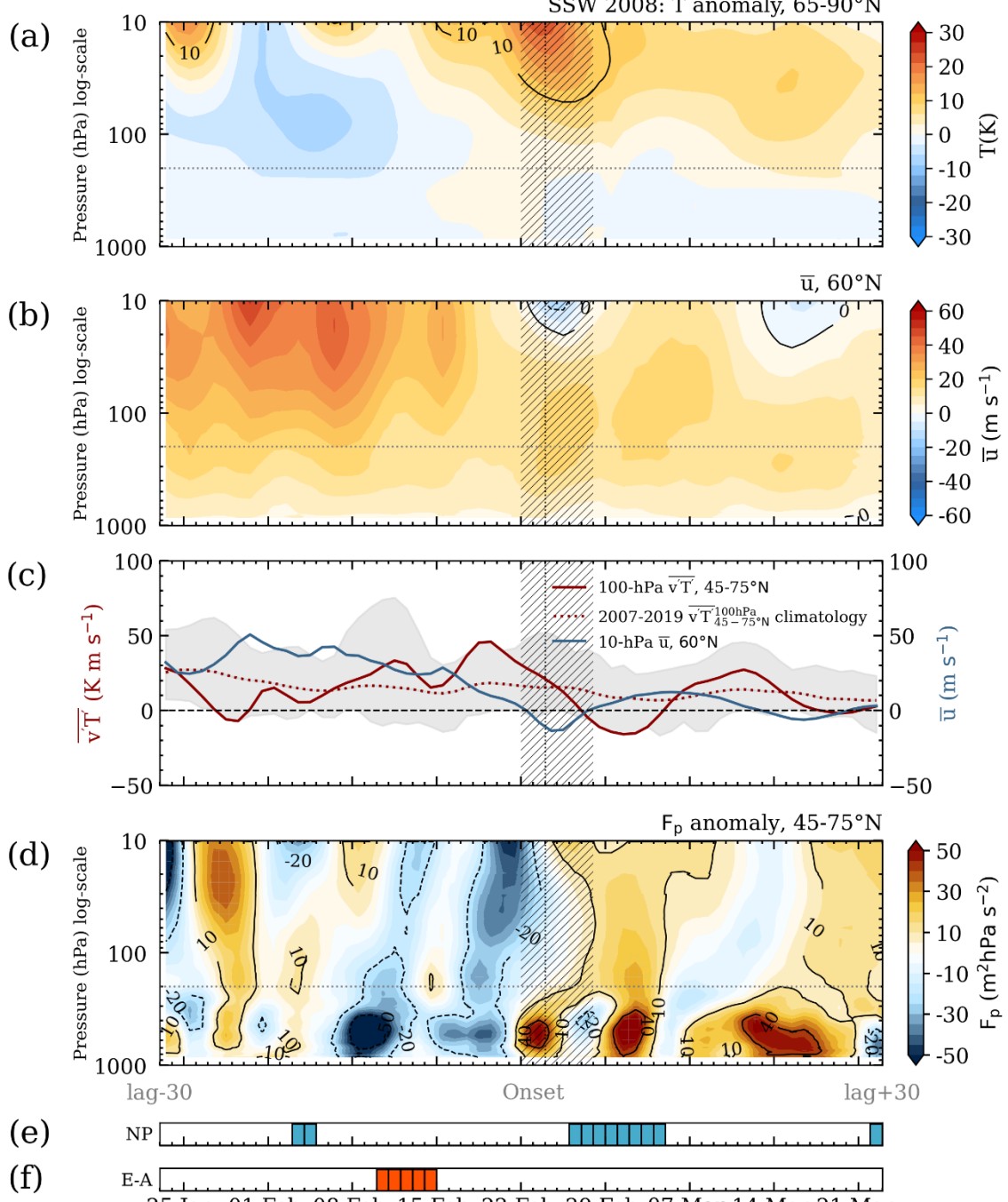

**Figure 3:** Time-height evolution of (a) area-weighted temperature anomaly averaged over 65-90° N, (b) zonal wind at 60° N, (c) 100 hPa eddy meridional heat flux averaged over 45-75° N (red solid line), its daily climatology (red dotted line) and zonal-mean zonal wind at 60° N and 10 hPa (blue solid line). Grey shading covers the region between daily minimum and maximum of the heat flux for the period 2007-2019. Time-height evolution of (d) anomaly of vertical component of Eliassen–Palm flux averaged over 45-75° N, (e) blocking index for the North Pacific region, (f) blocking index for the Euro-Atlantic region. Hatched region indicates dates when the zonal-mean zonal wind at 60° N and 10 hPa is negative, and the vertical dotted line indicates the day, when the polar (80-90° N) temperature anomaly reaches its maximum, i.e., the start of SSW recovery phase. The dotted horizontal line indicates 200 hPa (an approximate level of extratropical tropopause). The time interval is shown for +/- 30 days from the central date (22 February) of the 2008 SSW.

The negative $\overline{v'T'}$ peak from 27 February to 6 March coincided with the occurrence of blocking in the North Pacific shown in Fig. 3e. Note that for the 2007 SSW event, there was also a negative $\overline{v'T'}$ peak concurrent with the development of the North Pacific blocking (Figs. A2c, e). In Figure 2b, the manifestation of North Pacific blocking is evident through a positive 500 hPa geopotential height anomaly from 27 to 29 February. Notably, the arrangement of the 500 hPa geopotential height anomaly field corresponded to that of the 50 hPa field. The barotropic low-pressure system over eastern Eurasia indicated a polar vortex shift toward Eurasia. This alignment underscores a connection between stratospheric and tropospheric conditions.

For a more detailed analysis of the vertical propagation of wave activity and its impact on the circulation, the meridional cross sections of the divergence and vectors of the three-day averaged EP flux and $\overline{u}$ for the main phases of the 2008 SPV are shown in Figs. 2c, d. Throughout the SPV's displacement and disruption phase, there was a notable enhancement of EP flux convergence and the upward propagation of EP flux vectors. As the EP flux progressively propagated into the stratosphere and northward of 75° N, it resulted in a slowing down of the stratospheric $\overline{u}$ at the pole. From 21 to 23 February, it can be observed that easterly winds are already present in the upper stratosphere, while westerly winds prevail in the middle and lower stratosphere. According to Perlwitz and Harnik (2003) and Kodera et al. (2008), this negative wind shear indicates favourable conditions for the reflection of upward propagating wave packets. When these wave packets encounter a transition from lower regions where westerly winds support their upward propagation to easterly winds that oppose it, an effective barrier to upward propagation is formed and results in the reflection of part of the wave energy. Between 27 and 29 February, the downward propagation of the EP flux can be observed together with an acceleration of $\overline{u}$ in the stratosphere, indicating that wave reflection is taking place. During these days, it can also be observed that the EP flux divergence in the stratosphere from 10 to 70 hPa is maximised between 70° N and 80° N. According to Shaw et al. (2010), the presence of localized positive EP flux divergence can act as an indicator of a reflecting condition within the atmosphere. A similar EP flux evolution is observed for the 2007 event (Figs. A1c, d).

In addition to the EP flux analysis, to further examine the evidence for the relationship between the downward propagation of wave activity and the North Pacific blocking, we analyzed the evolution of the 3D Plumb flux (Fig. B1). Starting from 21-23 February (Fig. B1b), and particularly on 27-29 February 2008 (Fig. B1c), a downward propagation of wave activity is observed between about 250°E and 300°E along with a trough centred over 300°E and a positive barotropic geopotential height anomaly between 150°E and 200°E (North Pacific). This is observed together with the eastward tilt of the trough, implying downward propagation of the Rossby waves, aligning with the findings of Kodera et al. (2008). This in turn induced the formation of the North Pacific ridge, which then led to the formation of the North Pacific blocking.

The above analysis describes the main characteristics of the 2008 and 2007 SSW events as consistent with the characteristics of reflecting SSWs, which in turn is consistent with the findings of Kodera et al. (2008; 2016). On this basis, we classify these SSW events as reflecting.

### 4.2 Type II: SSW events from 2009 to 2019

One of the main features of the SSW events that occurred between 2009 and 2019 was the SPV split. The 2010, 2013, 2016 and 2019 events, respectively, were of mixed type, in which the vortex displaces and then splits (Figs. 4a, A5a, A7a, A9a). The 2009 and 2018 SSW events were of the split type (Figs. A3a, A11a). Figure 4a shows the SPV evolution of the 2019 event. A displacement of the vortex towards Eurasia was initiated around mid-December 2018. Beginning on 22 December, the displaced vortex elongated towards the North Atlantic. On 2 January, the vortex reversed and split into two lobes and then continued to split apart until the middle of January.

In the weeks following the vortex displacement in December 2018, the polar cap temperature anomaly in the stratosphere increased significantly and exhibited a marked downward propagation, extending to approximately 200 hPa (Fig. 5a). The long-lasting deep warming is another commonality of the 2009-2019 SSW events (Figs. A2a, A4a, A6a, A8a, A10a, A12a). Consistent with the polar cap temperature variability, the reversal of $\overline{u}$ lasted several weeks in these years, which contrasts with the short duration of the SPV reversal in the 2007-2008 events. The only exception was the 2016 SSW event, during which the SPV did not undergo a recovery phase as it was a final warming event. Nevertheless, this event, with its major SSW characteristics, merits inclusion in our analysis and it has also been the focus of previous studies such as Manney and Lawrence (2016).

Another interesting feature of these events is the prolonged and gradual wave activity propagation into the stratosphere. The evolution of $\overline{u}$ and $\overline{v'T'}$ in 2019 is shown in Fig. 5c. The $\overline{v'T'}$ peaked on 22 December during the displacement of the vortex, after which $\overline{u}$ began to weaken. Subsequently, as January commenced, $\overline{v'T'}$ peak underwent a gradual reduction. This was followed by an acceleration of $\overline{u}$ and the eventual recovery of the SPV. Similarly, in 2010, 2013 and 2016, the $\overline{v'T'}$ peaked during the vortex displacement and gradually decreased during $\overline{u}$ reversal (Figs. A6c, A8c, A10c). In these events, several successive and overlapping peaks in $\overline{v'T'}$ are responsible for the continuous and long-lasting upward wave propagation (about 40 days). In the split events of 2009 and 2018, the peaks of $\overline{v'T'}$ were enhanced before the vortex split and lasted for 38 and 24 days, respectively (Figs. A4c, A12c). They then gradually decreased until the vortex recovered.

In the time-height view of $F_p$ for the 2019 event (Fig. 5d), the enhanced wave activity pattern appears first from around 4 to 13 December in the troposphere below 300 hPa. This is followed by mildly negative $F_p$ in the whole atmospheric column. Around 23 December, negative $F_p$ peaked in the stratosphere extending vertically upward from about 200 hPa. This was preceded by tropospheric $F_p$ enhancement with a delay of about eight days. From 25 December, there was another wave activity pattern in the troposphere below 200 hPa, which continued until 1 January. After the reversal of $\overline{u}$, the troposphere featured a more divergent state, indicating a decrease of wave activity. In the 2009-2018 events, the negative $F_p$ peak in the stratosphere is preceded by a negative $F_p$ peak in the troposphere by a few days before either the SPV displacement (2010, 2013 and 2016) or the SPV split (2009 and 2018 events).

As for the tropospheric response, Euro-Atlantic blocking was observed in the troposphere during the SPV split from late December to mid-January, as shown by the positive 500 hPa geopotential height anomaly over the Euro-Atlantic in Fig. 4b.

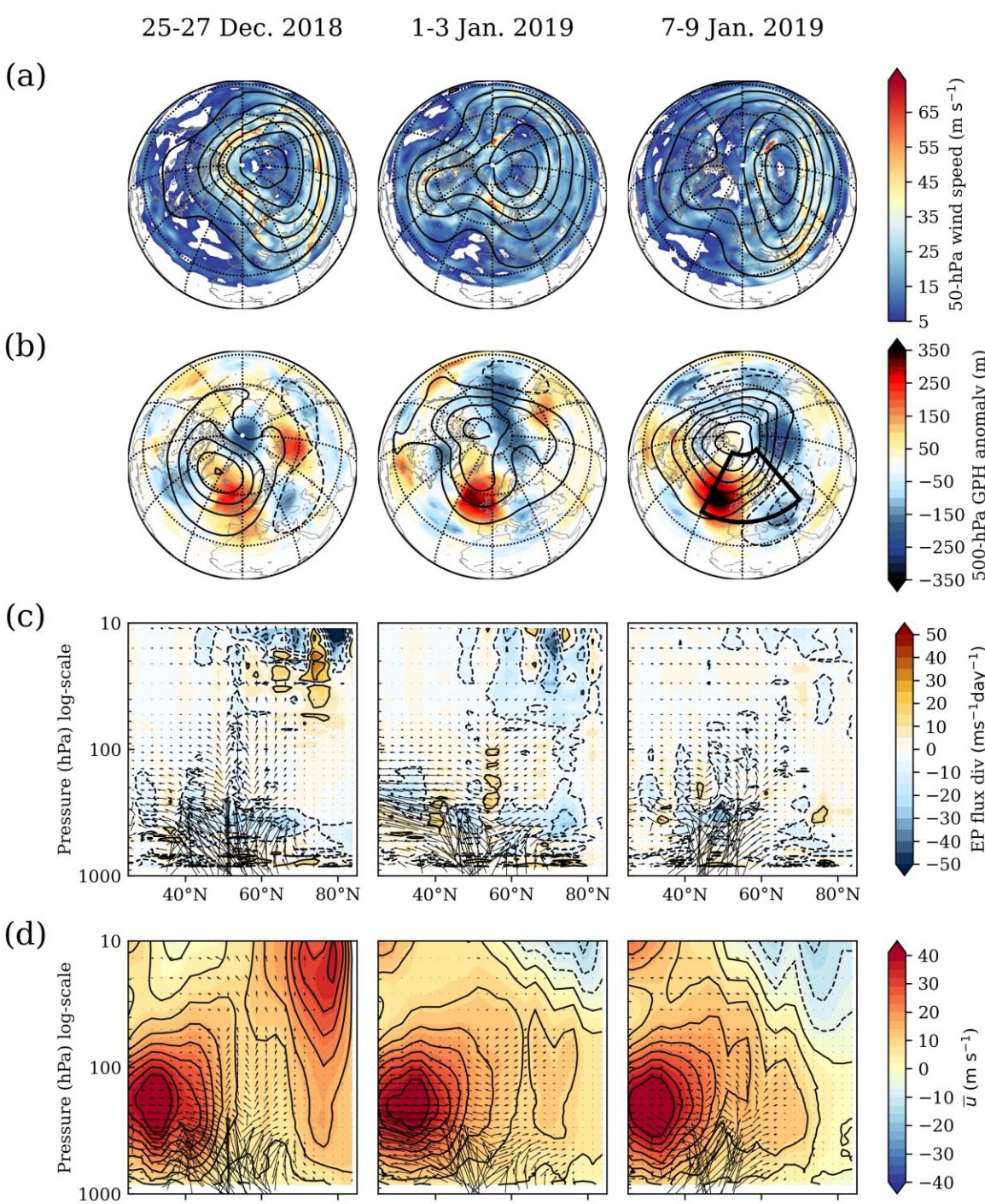

**Figure 4:** Same as for Fig. 2, but for the 2019 SSW. The three-day averaged parameters are shown for the three dates: during the SPV displacement (left), during the split (center) and after the SPV split (right). The black box indicates the selected Euro-Atlantic blocking region.

275

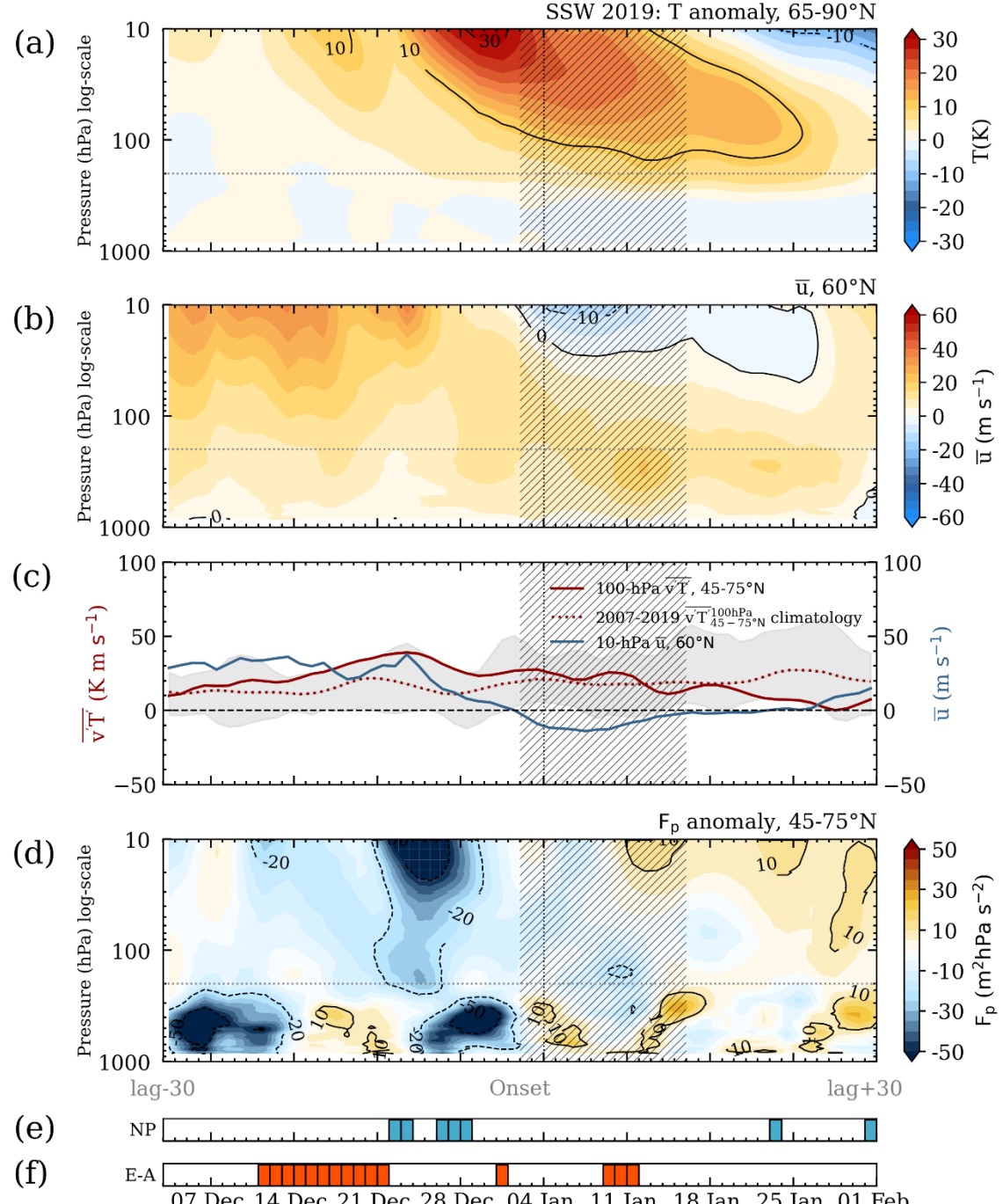

**Figure 5:** Same as for Fig. 3, but for the 2019 SSW. The time interval is shown for +/- 30 days from the central date (2 January).

The blocking index captured Euro-Atlantic blocking from 9 to 11 January in Fig 5f. It can also be observed that the Euro-Atlantic blocking coincided with the configuration of the positive 50-hPa geopotential height anomaly centered over the North Pole and extended towards the North Atlantic, suggesting the vertical stratosphere–troposphere connection (Fig. 4b). Butler et al. (2020) described that strongly positive stratospheric polar cap geopotential anomalies or the negative phase of the Northern Annular Mode (NAM) index were observed from the end of December until mid-January 2019, along with North Atlantic blocking. A possible link between this blocking and the SPV configuration in January 2019 is also suggested by Yessimbet et al. (2022b). Similarly, the 2009-2018 SSW events also

featured Euro-Atlantic blocking shortly after the SPV split (Figs. A4e-f, A6e-f, A8e-f, A10e-f, A12e-f), and the geopotential height configuration resembled a negative AO pattern (Figs. A3b, A5b, A7b, A9b, A11b).

A closer look at the EP flux during the first week of the recovery phase of the 2019 SSW reveals a continuous upward direction of wave propagation from the troposphere to the stratosphere, as shown in Figs. 4 c, d. The large EP flux convergence zone in the stratosphere in early January led to deceleration of $\overline{u}$. As the EP flux vectors propagated poleward and upward, $\overline{u}$ at the pole became easterly and extended downward. To further investigate the relationship between the North Atlantic blocking and the details of wave activity propagation, the 3D Plumb flux evolution and vertically resolved geopotential height anomalies are shown in Fig. B2. Along with the onset of North Atlantic blocking formation, it can be observed that the wave activity is enhanced outward from the positive geopotential height anomalies centred between 50°W and 0°. This suggests that wave packets originating from the North Atlantic block propagate into the stratosphere, thereby contributing to vortex weakening and further SSW development. Overall, the continuous upward propagation of the wave activity (shown by EP flux) into the stratosphere and a deep downward descent of a reversed $\overline{u}$ from the stratosphere into the troposphere during the SSW recovery are also typical for the 2009-2018 events (A3c-d, A5c-d, A7c-d, A9c-d, A11c-d).

The characteristics of the SSW events that occurred between 2009 and 2019 align with the description of absorbing SSW events as outlined by Kodera et al. (2016). On this basis, we classify the SSW events from 2009 to 2019 as absorbing.

### 4.3 Time-height view of the wave activity for 2007-2019 SSW events

Figure 6 displays the time-height evolution of the $F_p$ anomaly, averaged over 45-75° N, along with the blocking index within a +/- 30 day timeframe relative to each of the SSW events between 2007 and 2019.

Before each SSW, there was a pronounced increase in wave activity (indicated by blue shading) in the stratosphere, often extending below 100 hPa. In almost all events, an increase in stratospheric wave activity was preceded by an increase in tropospheric wave activity by several days, indicating an upward wave activity propagation. The exception was the 2007 event, during which the stratospheric enhancement occurred without a strong signal of tropospheric wave activity enhancement preceding it. However, approximately 20 days before that, there was another notable intensification of wave activity initially observed in the troposphere and then in the stratosphere above 70 hPa, which suggests the preconditioning of the SPV.

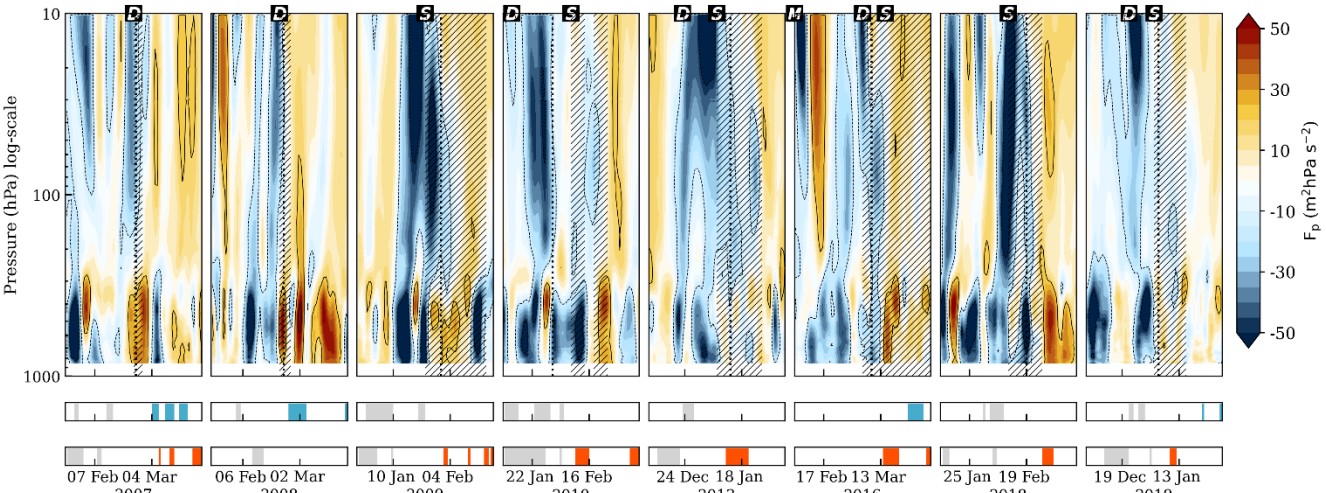

**Figure 6:** Time-height evolution of the anomaly of the vertical component of the Eliassen–Palm flux (45-75° N) within a +/- 30 day timeframe relative to each of the SSW events from 2007 to 2019. Hatched regions indicate dates when the zonal-mean zonal wind at 60° N and 10 hPa is negative, and the vertical dotted lines indicate the start of SSW recovery phase. Letters D and S indicate the approximate start of the SPV displacement and split, respectively. Letter M indicates the minor SSW event. The lower panels show the blocking index for the North Pacific (blue) and Euro-Atlantic region (orange). The blocking events before the SSWs are greyed out.

Comparing all individual events, the upward wave propagation signals associated with SSW were less pronounced and shorter in 2007 and 2008, as was the duration of $\overline{u}$ reversal in these years. In the other six events, the wave activity propagation signals prior to $\overline{u}$ reversal were more pronounced and longer.

The strongest signal of increased wave activity shortly before the SPV split was observed in 2009. For 2010, 2013 and 2019, there was an increase in $F_p$ in the two weeks prior to $\overline{u}$ reversal, coinciding with the SPV displacement. At the end of January and beginning of February 2016, there was an intensification of the $F_p$, which occurred during a minor SSW event, which was reported by Dörnbrack et al. (2018). In 2018, there were three episodes of $F_p$ enhancement, each separated by an interval of about two weeks before the split of the SPV. The first enhancement was continuous throughout the atmospheric column, with a few days lag between the troposphere and stratosphere anomalies. The second enhancement was observed in the troposphere and then in the stratosphere above 100 hPa, with no vertical continuity between the two regions. These two signals were not sufficient to weaken the vortex and trigger an SSW event. The third $F_p$ anomaly intensification signal was the strongest, with clear signs of upward wave propagation before the SPV split. Interestingly, a closer look at the 2018 SSW event (Fig. A12) reveals that only vertically continuous signals (marked by 100 hPa $\overline{v'T'}$ peaks) coincide with $\overline{u}$ weakening. The same is true for all other SSW events, which may indicate that the propagation or amplification of wave activity in the lower stratosphere (around 100 hPa) is more important for the weakening of SPV than its amplification in the middle and upper stratosphere.

Regarding the blocking events, in both the 2007 and 2008 events, the North Pacific blocking became apparent shortly after the $\overline{u}$ recovery. For the other SSW events, we find that the Euro-Atlantic blocking is observed immediately after the onset of the SSW recovery and during and after (for the 2018 event) the reversal of $\overline{u}$.

To better understand the differences between the SSW events, we also analyzed the evolution of the zonally averaged polar (75-90° N) static stability anomalies (Fig. 7). In the 2009-2019 SSW events, an increase in the static stability anomaly near 300 hPa can be observed

as $\overline{u}$ reverses. This static stability enhancement indicates a strengthening of the tropopause inversion layer (TIL) in the polar region in the aftermath of SSWs. This observation confirms the findings of Grise et al. (2010), who demonstrated that the magnitude of the TIL is enhanced following SSWs. Also, the case studies of Wargan and Coy (2016) using reanalyses and of Wang et al. (2016) using Formosat3/COSMIC RO measurements described enhancement of the static stability in the vicinity of the polar tropopause following the 2009 SSW event.

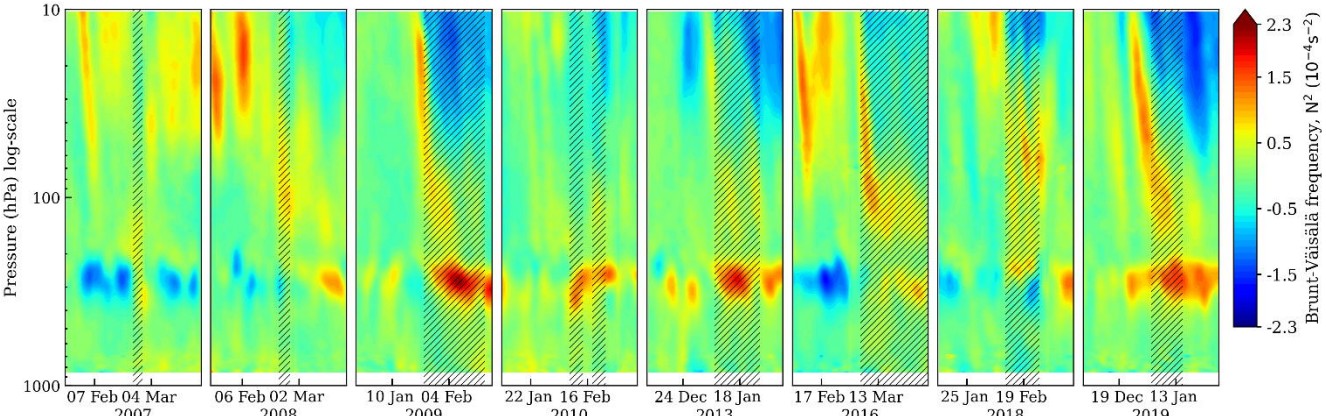

**Figure 7:** Time-height evolution of the anomalous static stability or the Brunt Väisälä frequency, $N^2$, (75-90° N) within a +/- 30 day timeframe relative to each of the SSW events from 2007 to 2019. Hatched regions indicate dates when the zonal-mean zonal wind at 60° N and 10 hPa is negative.

In our observations, the 2009 and 2013 SSW events had the strongest enhancement of the polar TIL magnitude. We also note the descending enhancement of static stability from the stratosphere to the tropopause level during the onset of the SSWs, which is observed in the static stability anomalies for the 2009, 2016, 2018, and 2019 SSWs and in its absolute values for all SSWs (Fig.S4a). This shows that the absorbing SSW events 2009-2019 had stronger and more prolonged impact (in terms of thermal heating) on the UTLS and the enhancement of the polar TIL than the reflecting events in 2007 and 2008. For the reflecting events, the magnitude of TIL enhancement is much weaker compared to the absorbing events. In 2008, the enhancement of static stability anomaly occurred in the late March during the final $\overline{u}$ reversal.

## 5 Discussion and Conclusions

The main objective of this study was to characterize the synoptic and dynamic conditions of SSWs and to investigate the link to blocking events from an observational perspective. We used GNSS RO observation for these analyses as the dataset resolves the relevant features to provide information on the stratosphere–troposphere coupling.

Within the timeframe of available RO data spanning from 2007 to 2019, we examined a total of eight major SSW events, including a final SSW event in 2016. To characterize SSWs, we analyzed RO temperature and geopotential height profiles on isobaric surfaces, also serving as a basis for deriving daily geostrophic winds and quasi-geostrophic EP fluxes. We also computed the blocking index to assess blocking events.

The results showed that the RO data resolves reasonably well all the main dynamic features of SSWs and troposphere-stratosphere coupling phenomena. While the geostrophic wind speed near the upper-tropospheric subtropical jet and at the SPV level may be slightly underestimated, as noted by Scherllin-Pirscher et al. (2014), our study showed that the SPV evolution is well captured. Analyzing the

evolution of the SPV, we classified the SSW events into distinct categories, specifically the displacement, split, and mixed-type events. The 2007 and 2008 SSW events were identified as displacement events, while the 2009 and 2018 events were classified as split events and 2010, 2013, 2016 and 2019 as mixed-type events. A case in point is the 2019 SSW event, which agrees with the findings of Lee and Butler (2020).

Furthermore, our study shows that the key patterns of quasi-geostrophic EP fluxes are well captured and consistent with established theory and the existing literature. Building on the analysis of EP flux evolution, we have classified the SSW events into two categories: reflecting and absorbing events. Thus, the 2007 and 2008 SSW events were categorized as reflecting, and the remaining events between 2009 and 2019 as absorbing. For the reflecting SSWs, our observations revealed a short duration of the $\overline{u}$ reversal, and a concurrent downward propagation of EP flux during the initial week of the SSW recovery phase. The analysis of 3D Plumb flux showed that the downward propagating wave packets induced a trough over eastern Canada and North America and the formation of a ridge over the North Pacific, leading to the onset of North Pacific blocking. On the other hand, absorbing SSW events exhibited a more prolonged $\overline{u}$ reversal and upward propagation of the EP flux. During the recovery phase, these events were accompanied by the formation of blocking in the Euro-Atlantic region and a geopotential height configuration resembling a negative AO pattern. Enhanced wave activity originating from the North Atlantic blocking was observed to propagate into the stratosphere, thereby potentially contributing to vortex weakening and further SSW development.

These observations agree with the findings of Perlwitz and Harnik (2004), who suggested that there are two types of stratospheric winter conditions, reflective and non-reflective, which are characterized by different downward dynamic interactions similar to those observed in our study.

Although the reflecting events of 2007 and 2008 were also classified as displacement events and the absorbing events of 2009-2019 as split and mixed events, the relatively limited number of SSW events studied does not allow statistical conclusions to be drawn concerning the consistent alignment of these two classification types. As highlighted in the Introduction, these classifications are rooted in distinct phases of an SSW. The classification based on polar vortex geometry is for the mature phase, whereas the classification based on the propagation of planetary wave activity is for the recovery phase of SSWs.

Nevertheless, we show that SSW reflecting/absorbing events differ in the magnitude of the downward impact (manifested e.g. in TIL variability, downward propagation of easterly wind and temperature anomalies) and correspond to specific divergent tropospheric responses. Reflecting events connected to vortex displacement are observed to trigger downward wave propagation inducing blocking over the North Pacific region while absorbing events connected to vortex split are associated with blocking over the North Atlantic and upward wave propagation. The magnitude of the downward impact may be one of the factors to consider in addressing the open question of whether displacement or split events trigger different responses in the tropospheric circulation.

Concerning the vertical structure of the quasi-geostrophic EP flux, we observed a consistent pattern of an enhanced upward EP flux ($F_p$) preceding the $\overline{u}$ reversal in each SSW event. We also observed evidence of upward propagation of wave activity, as prior to each SSW the intensification in $F_p$ in the troposphere it preceded that in the stratosphere. This aligns with the hypothesis that an increase in stratospheric wave activity is typically preceded by a burst in wave activity within the troposphere (Polvani and Waugh, 2004). Interestingly, this contradicts the results of Jucker (2016), who, based on an idealised general circulation model (GCM), did not observe any tropospheric enhancement of wave activity propagating into the stratosphere prior to SSWs. It is worth emphasizing that the peak of wave activity amplification exhibited distinct temporal characteristics between reflecting and absorbing SSW events. Specifically,

the amplification peak was shorter for reflecting SSWs than for absorbing events. This suggests the relevance of considering the time scales associated with wave activity pulses. This aspect agrees with the finding of Sjoberg and Birner (2012), who suggested that longer-duration wave activity pulses are more effective in generating SSWs than shorter yet stronger pulses.

In addition, the analysis of polar static stability anomalies showed that the SSWs are followed by then enhanced polar TIL, which was strongest for the 2009 and 2013 SSW events and weakest for the 2007 and 2008 events. This indicates that the strength of the TIL is influenced by the magnitude of the SSW and its downward impact. Given that TIL enhancement can further influence stratosphere-troposphere coupling and tropospheric circulation, this further emphasises the distinction between SSW events and their downward influences.

In conclusion, our findings underscore the applicability of GNSS RO for the exploration of atmospheric circulation dynamics. Due to its high vertical resolution, GNSS RO has the potential for studying the interplay between tropopause structure and wave activity propagation. A detailed study of the relationship between tropopause structure and wave activity propagation relevant to SSW events should be investigated in future GNSS RO studies.

**Data availability**

The Wegener Center (WEGC) GNSS RO record OPSv5.6 profiles are available at https://wegcwww.uni-graz.at/data-store/WEGC/OPS5.6:2020.1/.

**Author contributions**

KY contributed to the development of theory, performed the computational implementation and data analysis, produced all figures, and wrote the first draft of the paper. AKS provided the initial design for this study. FL provided the gridded RO data set. All authors contributed to the design of the study and figures, discussion of the results, and to the revision of the text. AKS, FL, AO provided guidance and advice on all aspects of the study.

**Competing interests**

The authors declare that none of them has any competing interests.

**Acknowledgements**

We thank WEGC EOPAC team for providing the OPSv5.6 data profiles. We are grateful to Barbara Scherllin-Pirscher and Lukas Brunner for providing the software used in our study to estimate the geostrophic winds and blocking indices (https://github.com/lukasbrunner/blocking), respectively. This work was funded by the Austrian Science Fund (FWF) under research Grant W1256 (Doctoral Programme Climate Change: Uncertainties, Thresholds and Coping Strategies). The authors also acknowledge financial support by the University of Graz. For the purpose of open access, the author has applied a CC BY public copyright license to any author accepted manuscript version arising from this submission.

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

**Appendix A: SSWS 2007, 2009, 2010, 2013, 2016 and 2018**


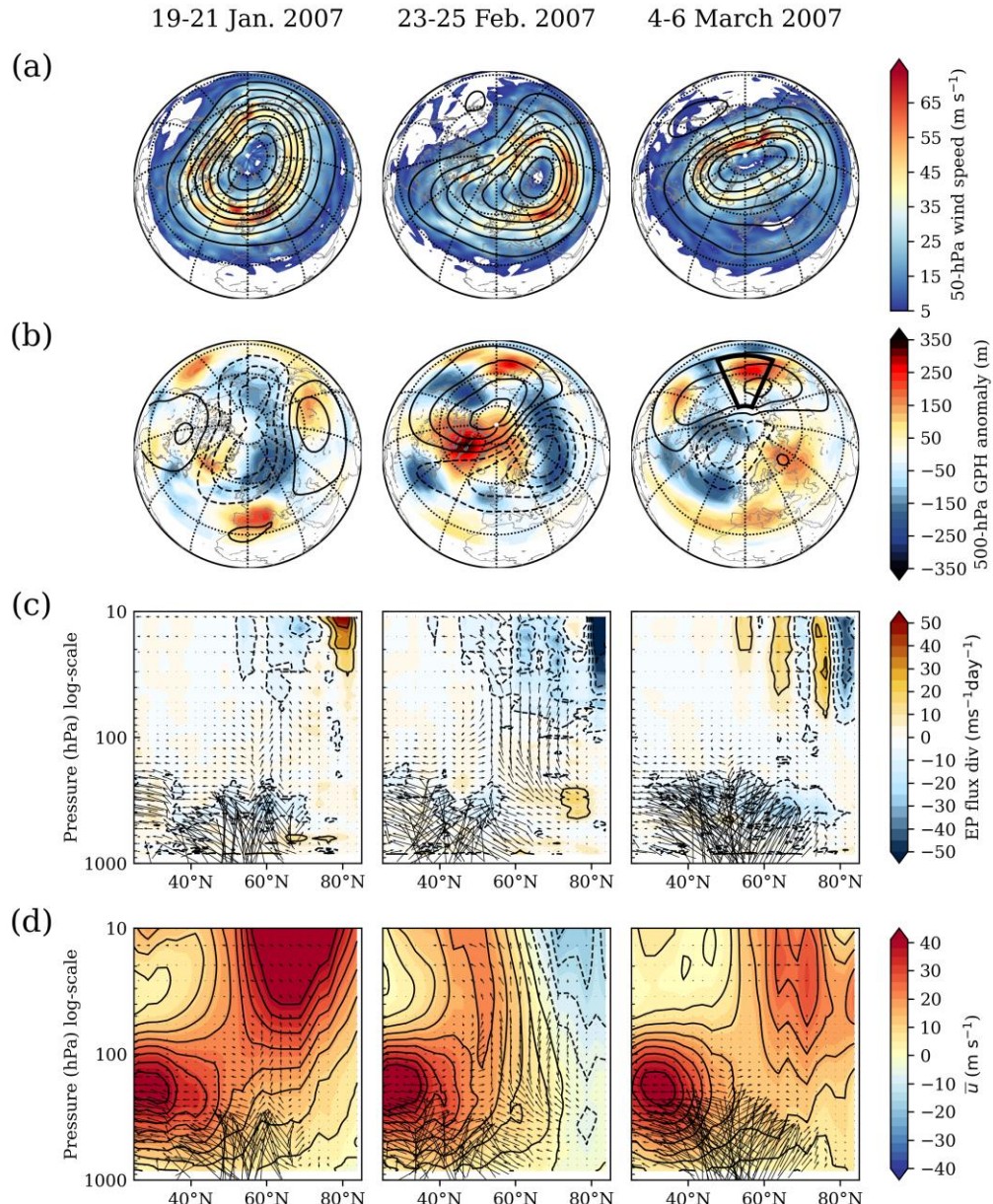

**Figure A1:** Evolution of the SSW 2007 shown for the three dates: before the SPV displacement (left), during its displacement (center) and during its recovery (right). (a) 50 hPa wind speed (shading) and 50 hPa geopotential height (contours). (b) 500 hPa geopotential height anomaly (shading) and 50 hPa geopotential height anomaly (contours). The black box indicates the selected North Pacific blocking region. (c) Meridional cross-sections of Eliassen–Palm flux vectors and divergence (shading). (d) Meridional cross-sections of Eliassen–Palm flux vectors and zonal wind (shading). The vectors are plotted for every 5th pressure level.

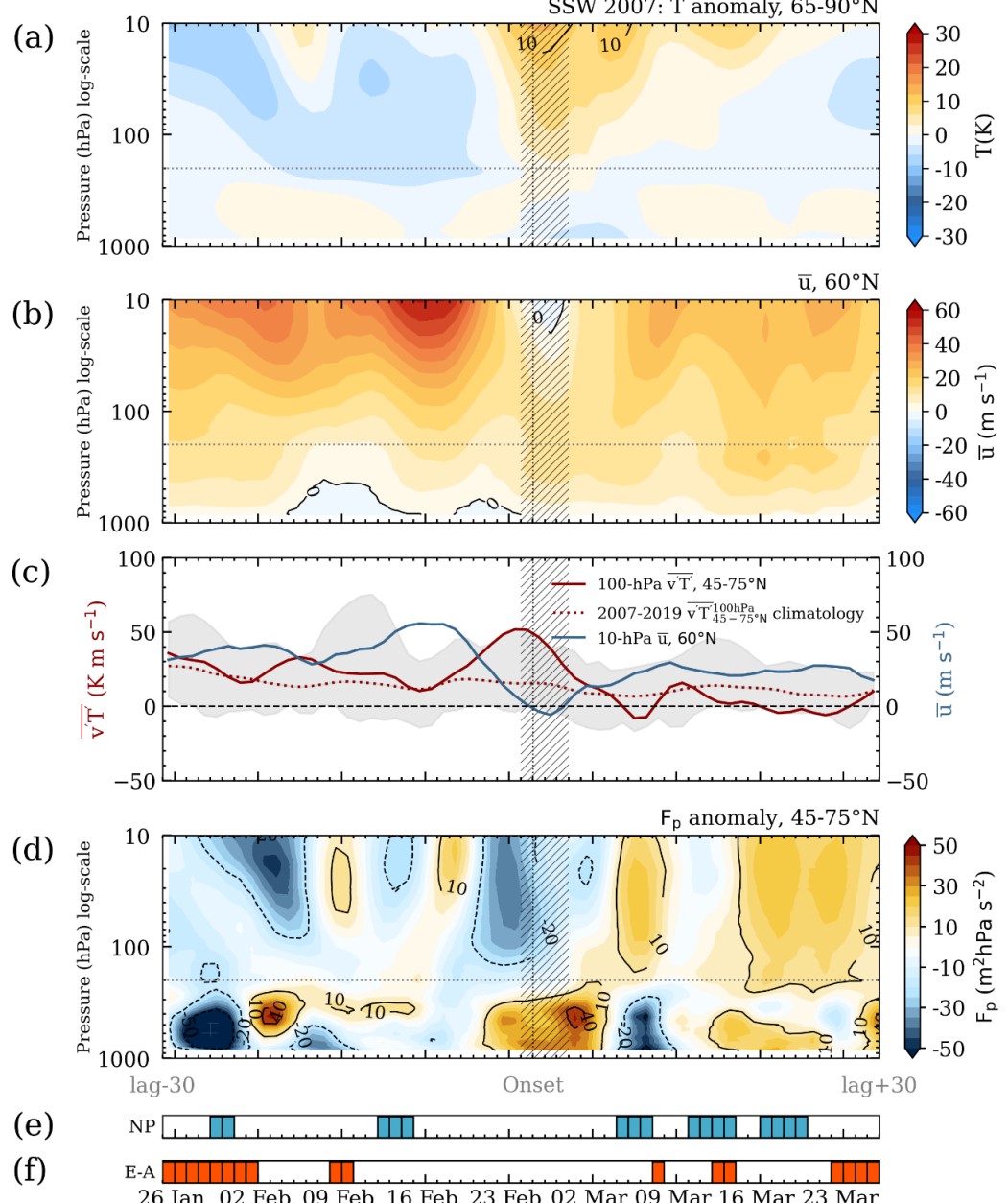

**Figure A2:** Time-height evolution of (a) area-weighted temperature anomaly averaged over 65-90° N, (b) zonal wind at 60° N, (c) 100 hPa eddy meridional heat flux averaged over 45-75° N (red solid line), its daily climatology (red dotted line) and zonal-mean zonal wind at 60° N and 10 hPa (blue solid line). Grey shading covers the region between daily minimum and maximum of the heat flux for the period 2007-2019. Time-height evolution of (d) anomaly of vertical component of Eliassen–Palm flux averaged over 45-75° N, (e) blocking index for the North Pacific region, (f) blocking index for the Euro-Atlantic region. Hatched region indicates dates when the zonal-mean zonal wind at 60° N and 10 hPa is negative, and the vertical dotted line indicates the day, when the polar (80-90° N) temperature anomaly reaches its maximum, i.e., the start of SSW recovery phase. The dotted horizontal line indicates 200 hPa (an approximate level of extratropical tropopause). The time interval is shown for +/- 30 days from the central date (24 February) of the 2007 SSW.

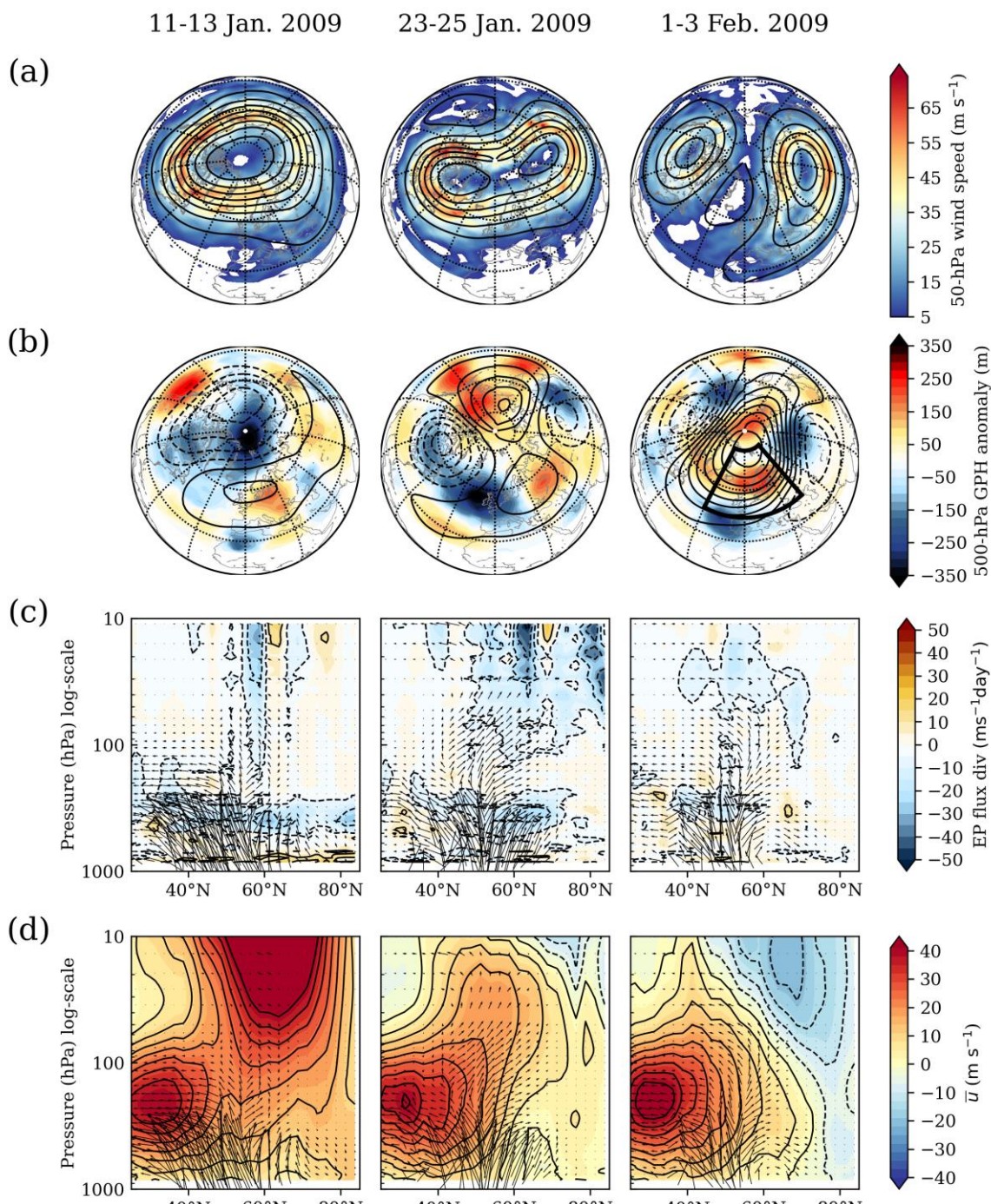

**Figure A3:** Same as for Fig. A1, but for the 2009 SSW. The three-day averaged parameters are shown for the three dates: before the SPV split (left), during the split (center) and after the split (right). The black box indicates the selected Euro-Atlantic blocking region.

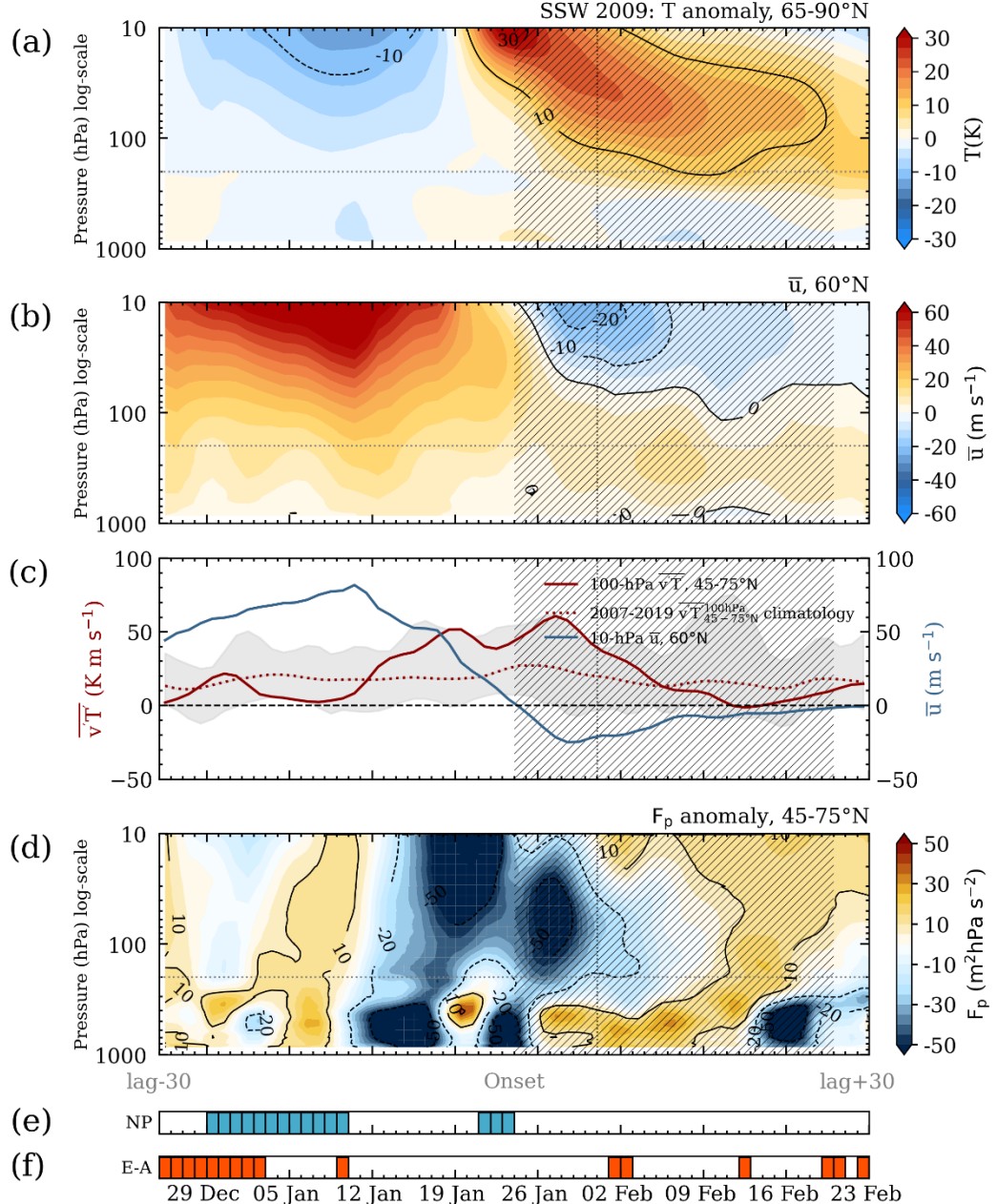

**Figure A4:** Same as for Fig. A2, but for the 2009 SSW. The time interval is shown for +/- 30 days from the central date (24 January).

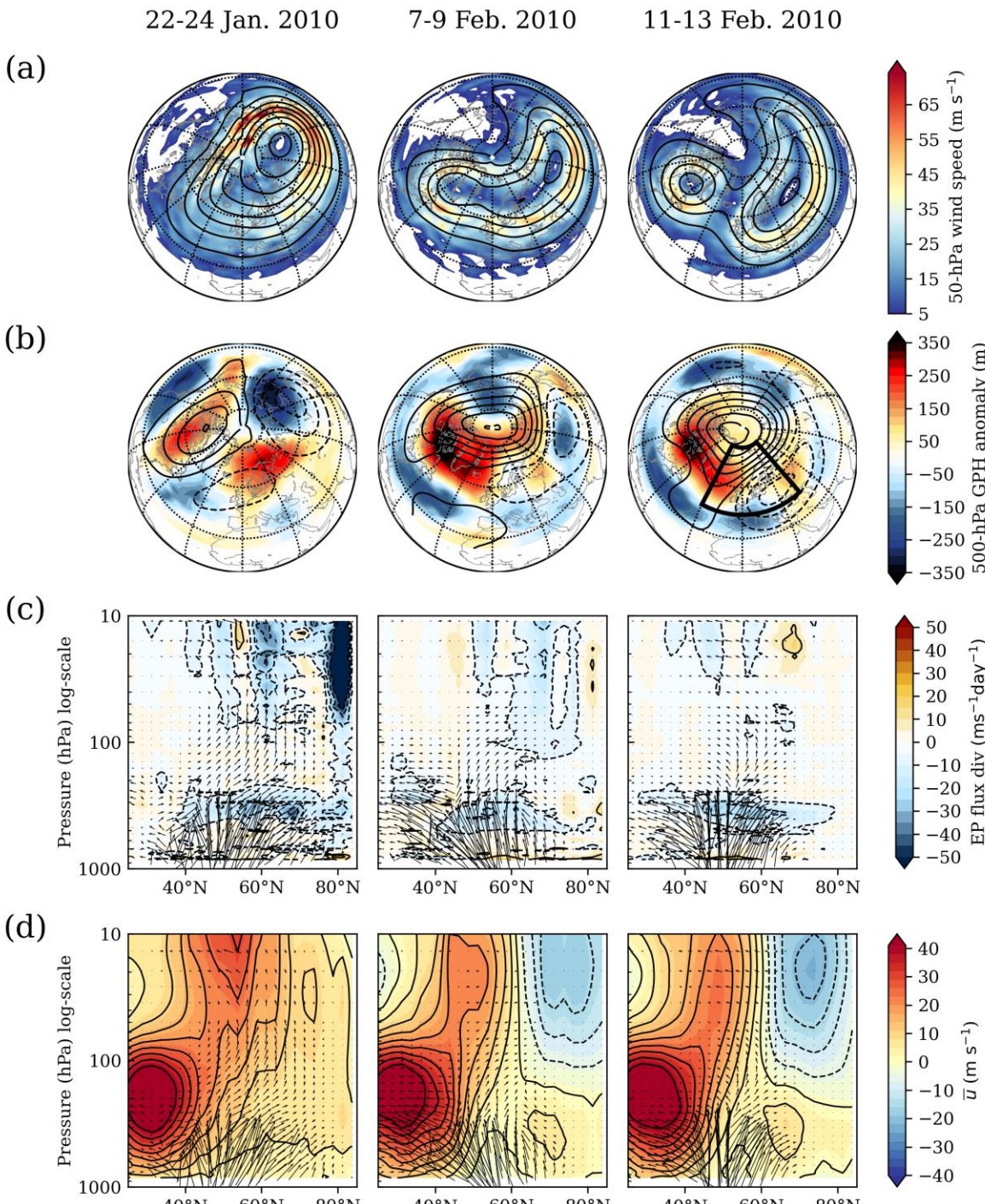

**Figure A5:** Same as for Fig. A1, but for the 2010 SSW. The three-day averaged parameters are shown for the three dates: during the SPV displacement (left), during the split (center) and after the split (right). Black box indicates defined Euro-Atlantic blocking region.

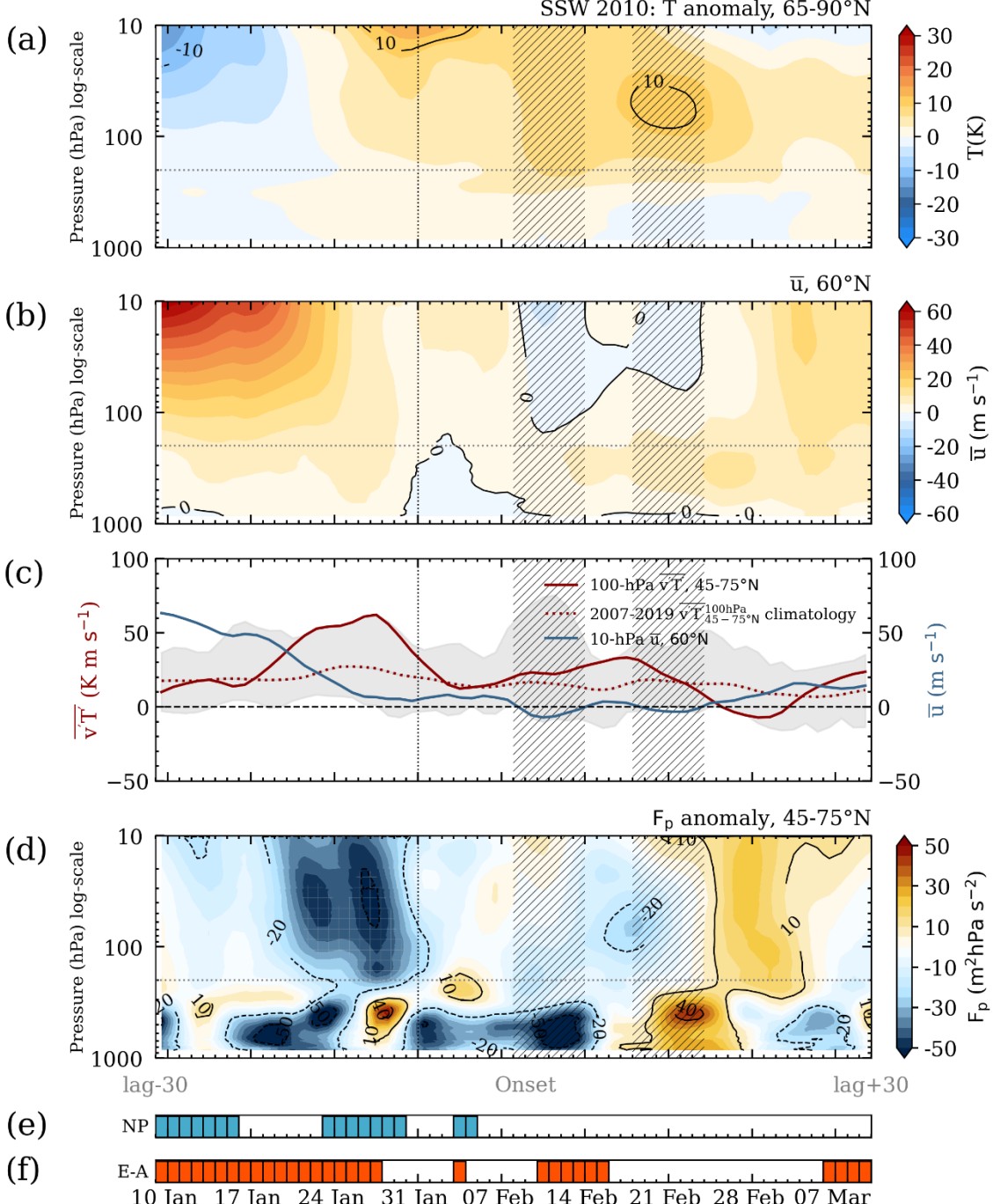

**Figure A6:** Same as for Fig. A2, but for the 2010 SSW. The time interval is shown for +/- 30 days from the central date (8 February).


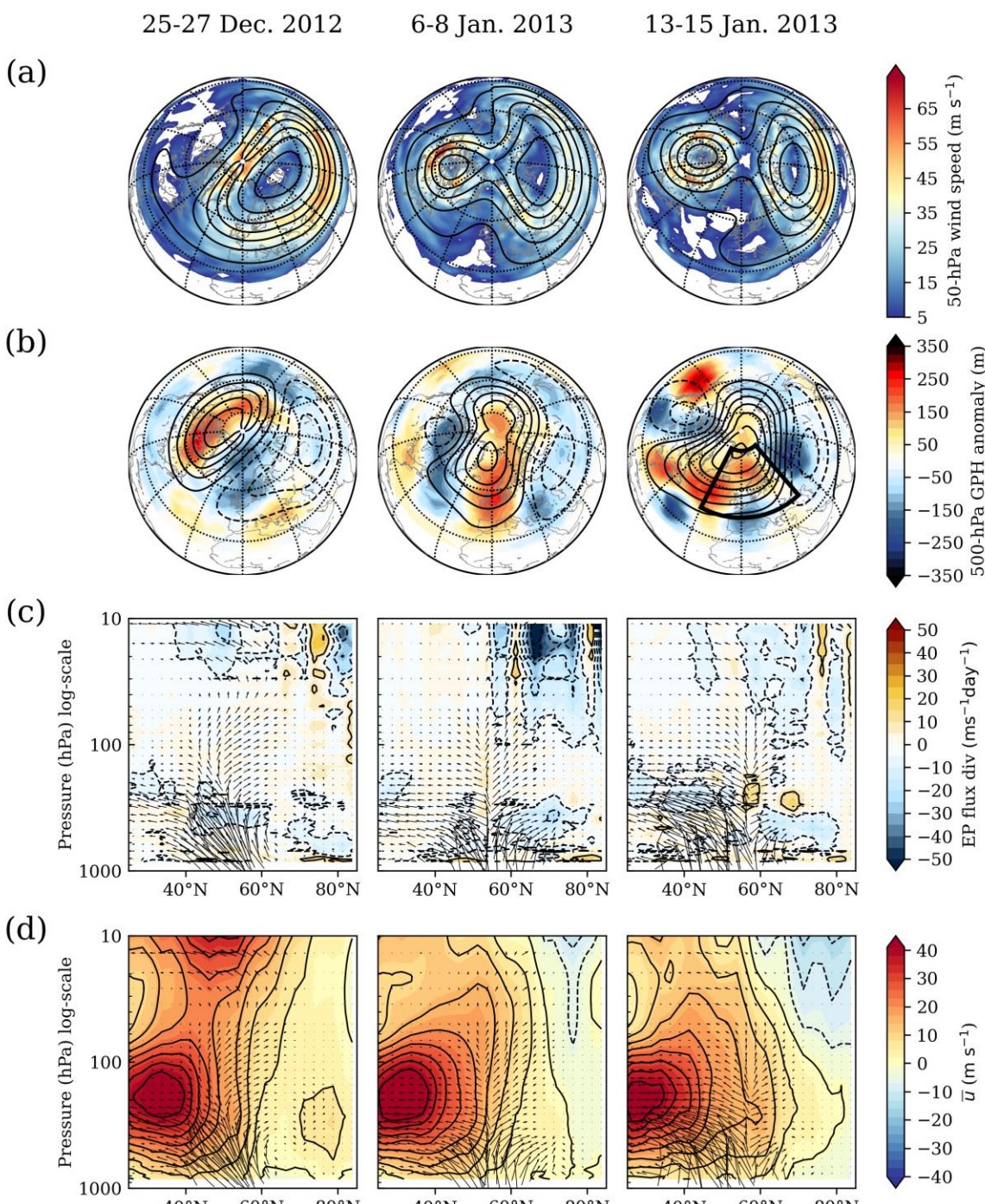

**Figure A7:** Same as for Fig. A1, but for the 2013 SSW. The three-day averaged parameters are shown for the three dates: during the SPV displacement (left), during the split (center) and after the split (right). Black box indicates defined Euro-Atlantic blocking region.


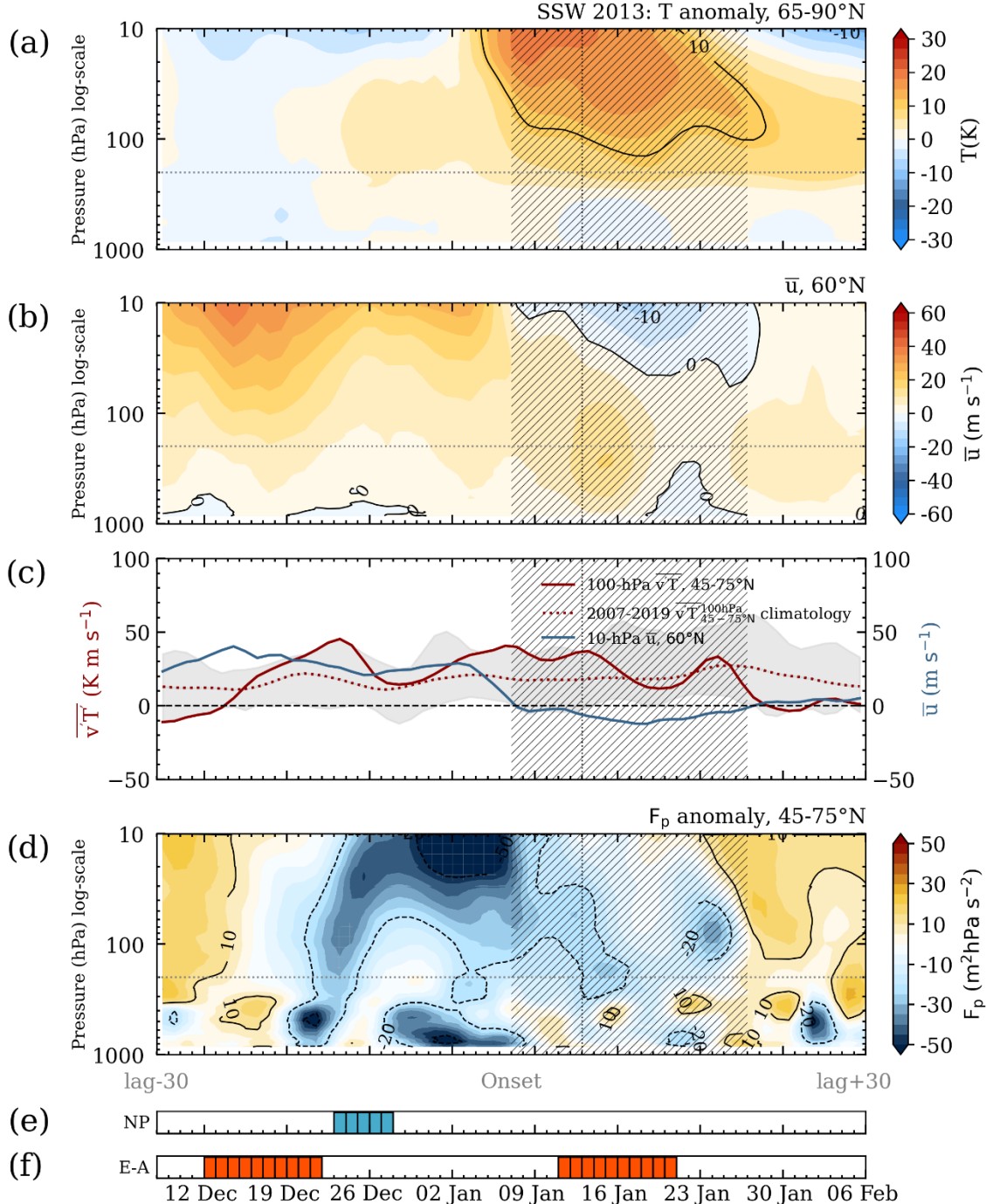

**Figure A8:** Same as for Fig. A2, but for the 2013 SSW. The time interval is shown for +/- 30 days from the central date (7 January).

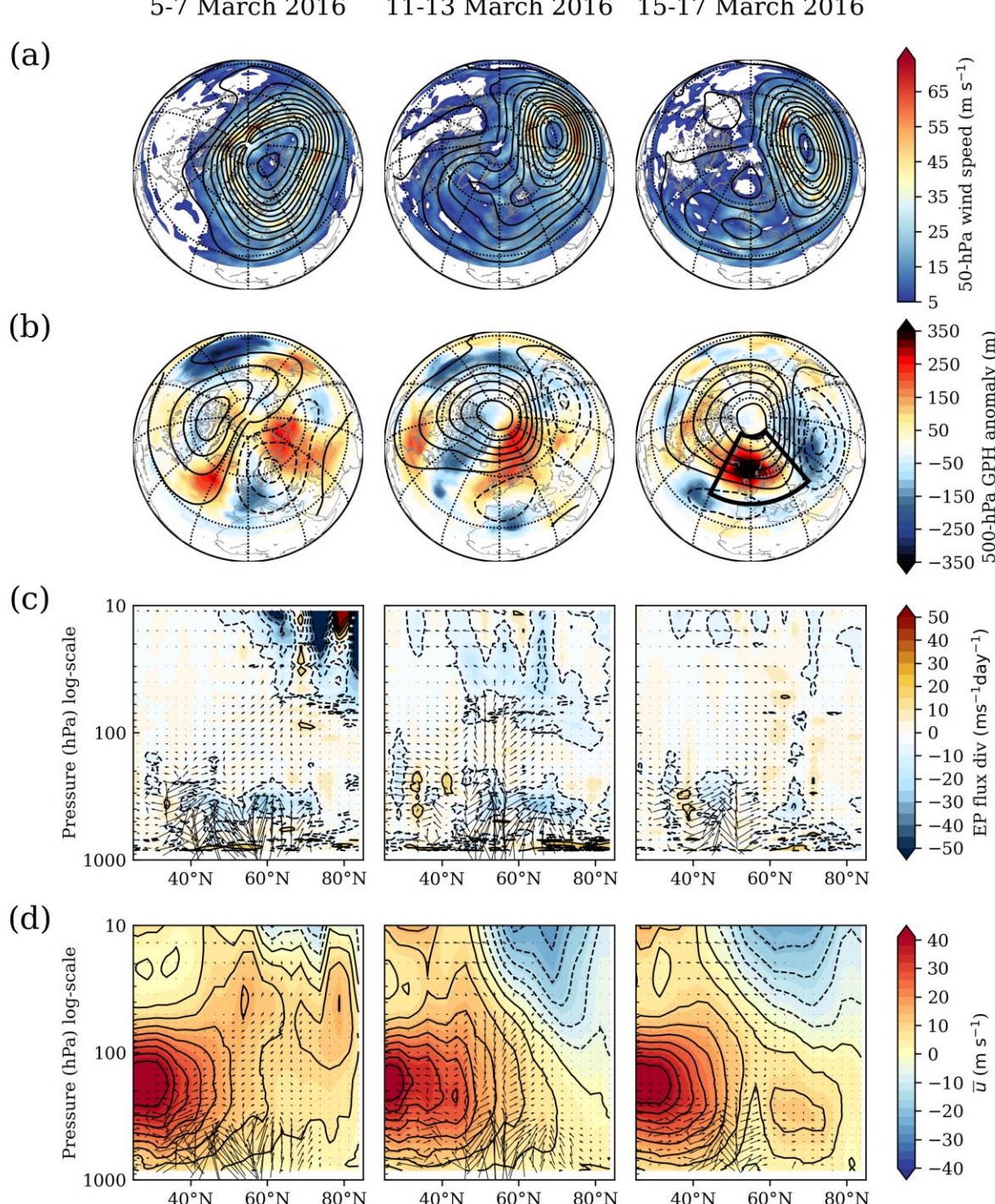

**Figure A9:** Same as for Fig. A1, but for the 2016 SSW. The three-day averaged parameters are shown for the three dates: during the SPV displacement (left), during the split (center) and after the split (right). Black box indicates defined Euro-Atlantic blocking region.


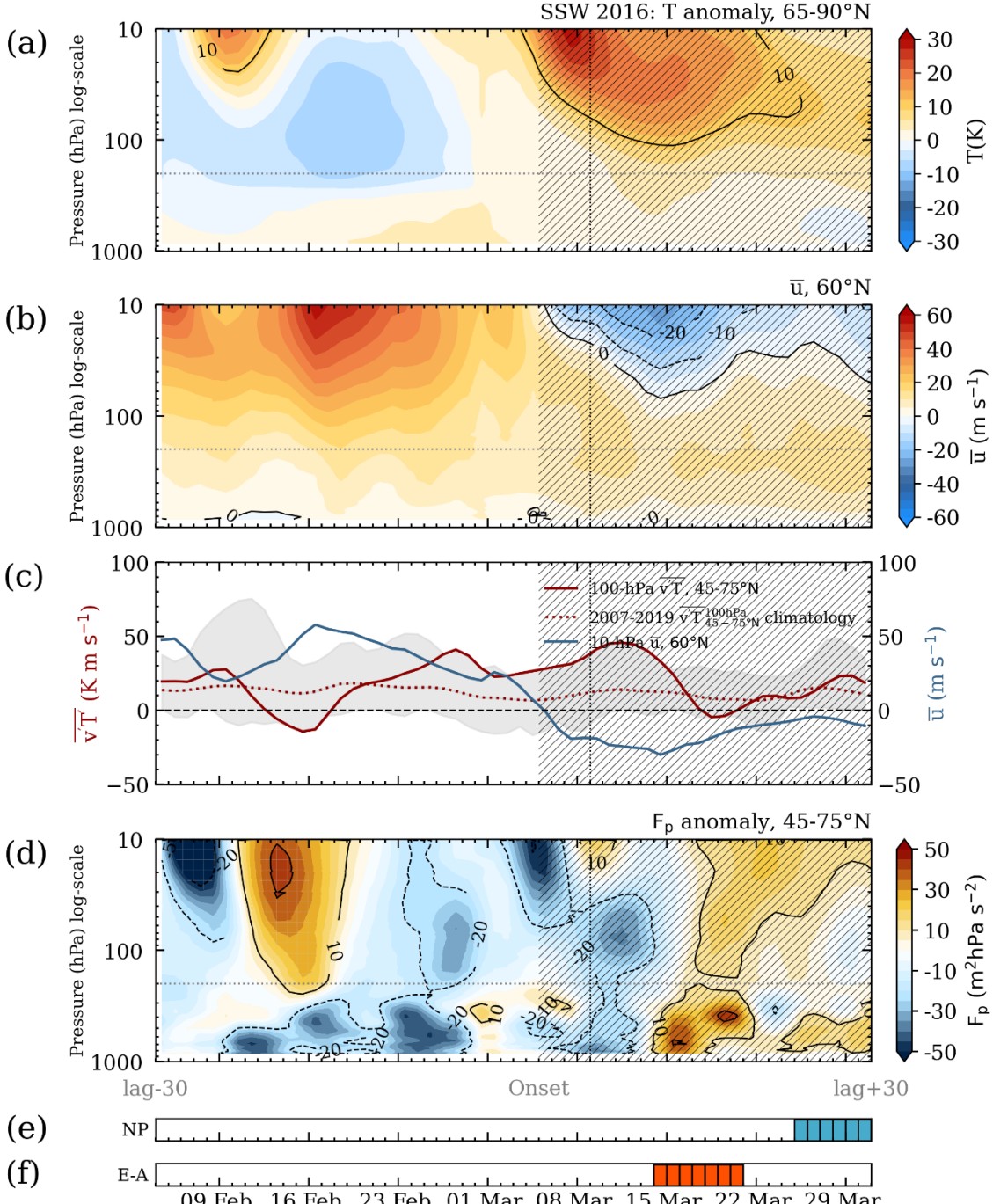

**Figure A10:** Same as for Fig. A2, but for the 2016 SSW. The time interval is shown for +/- 30 days from the central date (5 March).


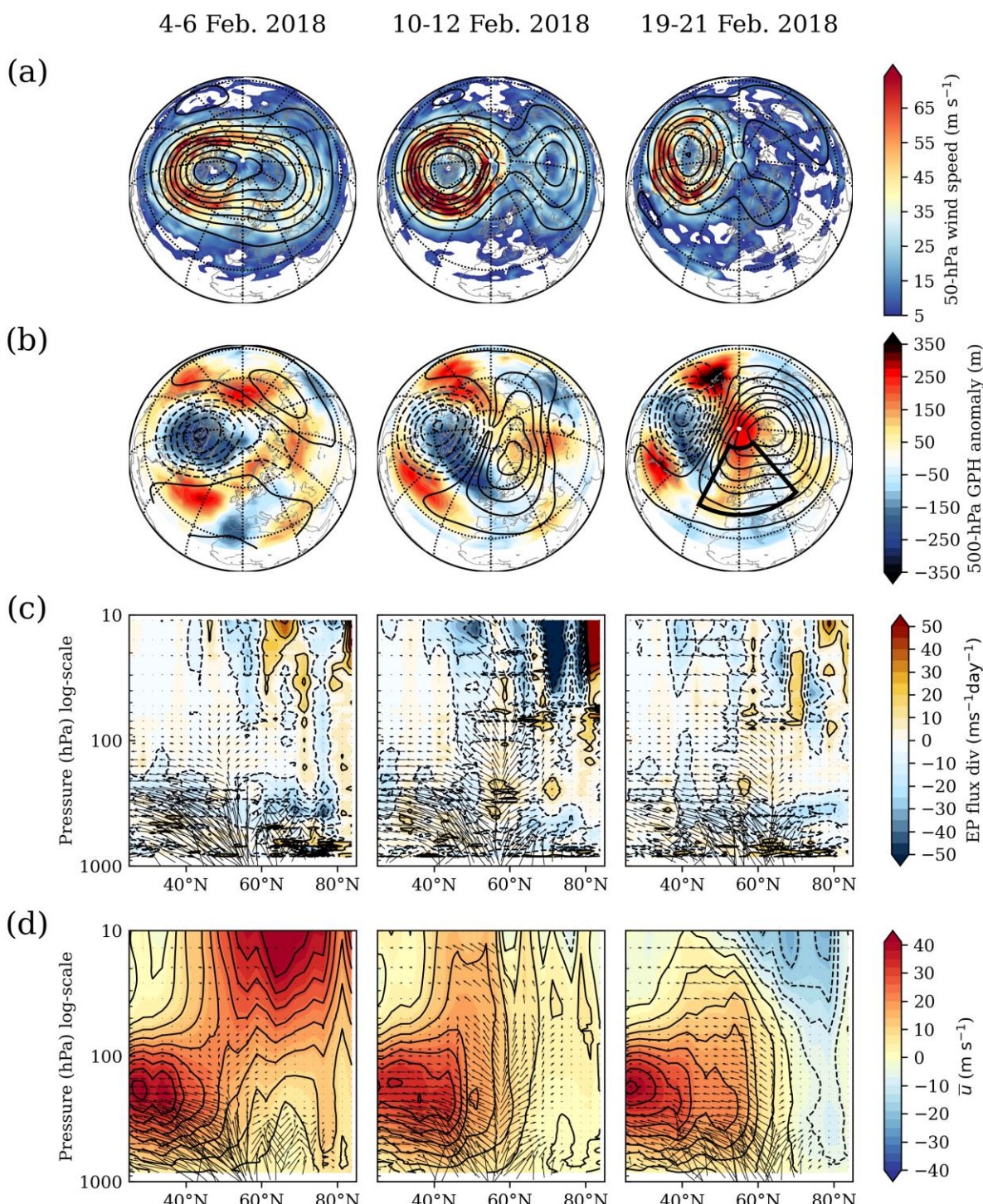

**Figure A11:** Same as for Fig. A1, but for the 2018 SSW. The three-day averaged parameters are shown for the three dates: before the SPV split (left), during the split (center) and after the split (right). Black box indicates defined Euro-Atlantic blocking region.

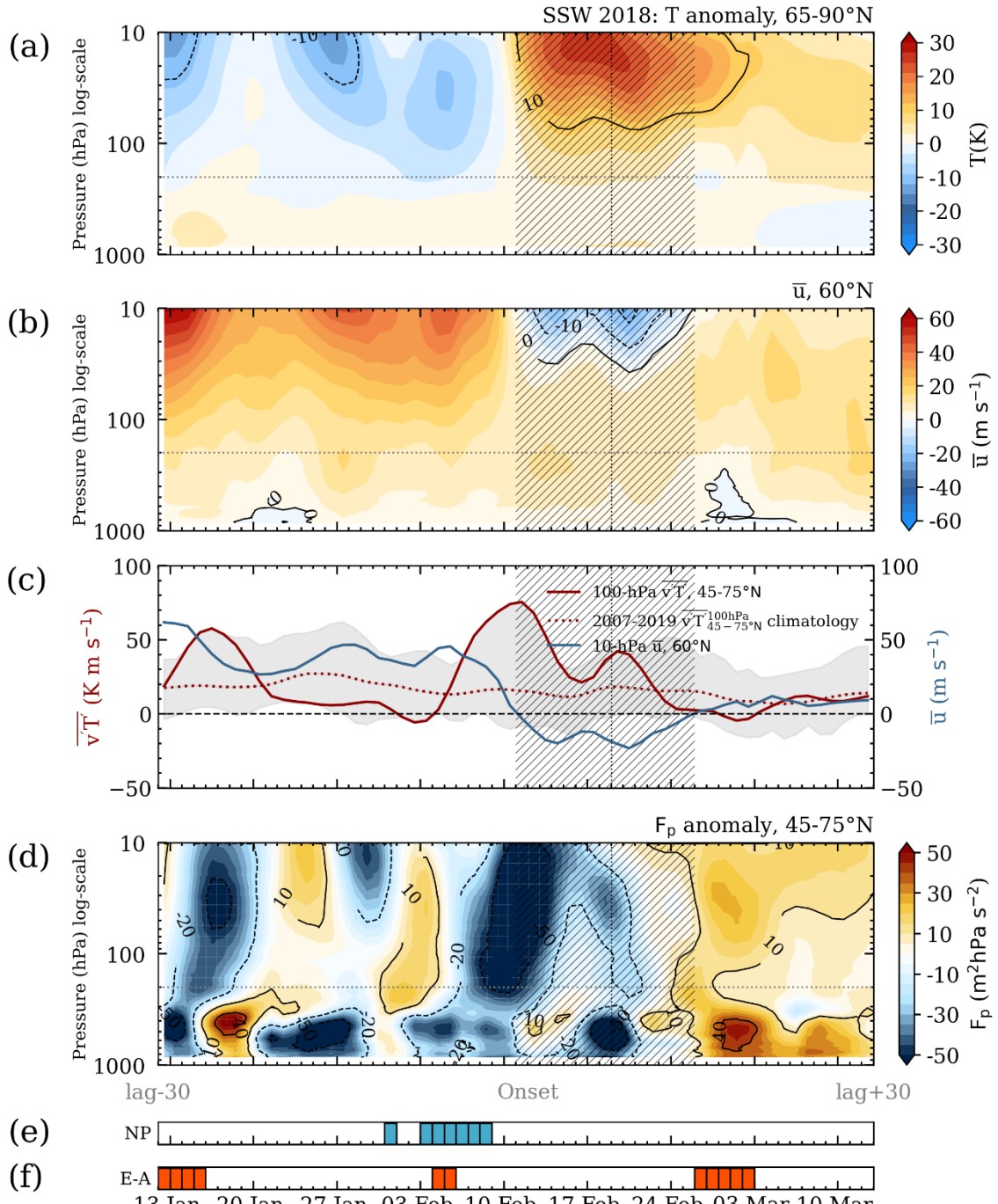

**Figure A12:** Same as for Fig. A2, but for the 2018 SSW. The time interval is shown for +/- 30 days from the central date (11 February).

**Appendix B: Plumb flux**

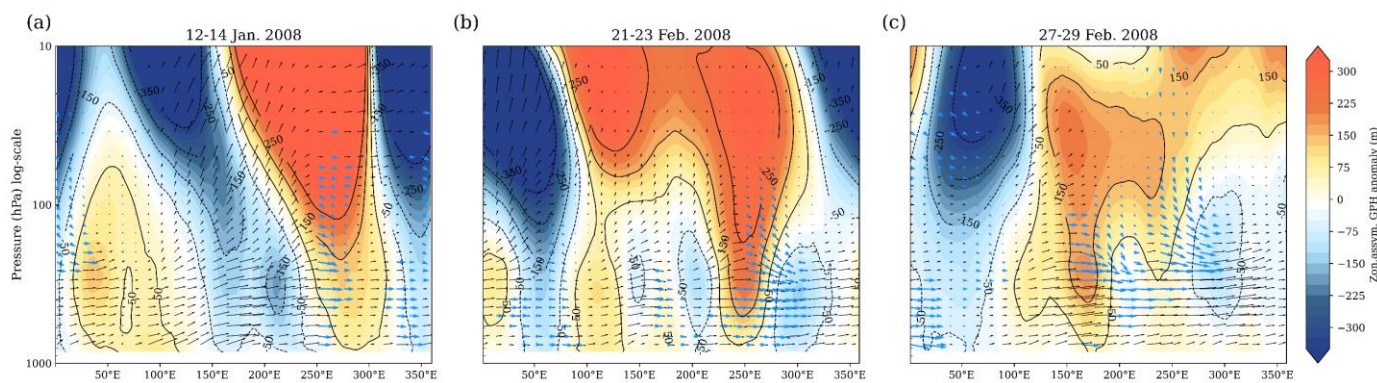

**Figure B1:** Evolution of the vertical and zonal components of the 3D Plumb flux (arrows), shown for the SSW 2008: before the SPV displacement (12-14 Jan.), during its displacement (21-23 Feb.) and during its recovery (27-29 Feb.) The vectors are plotted for every 5th longitude and pressure level. Blue vectors denote where the vertical component of Plumb flux is negative. Shading indicates the zonally asymmetric component of the geopotential height anomaly.

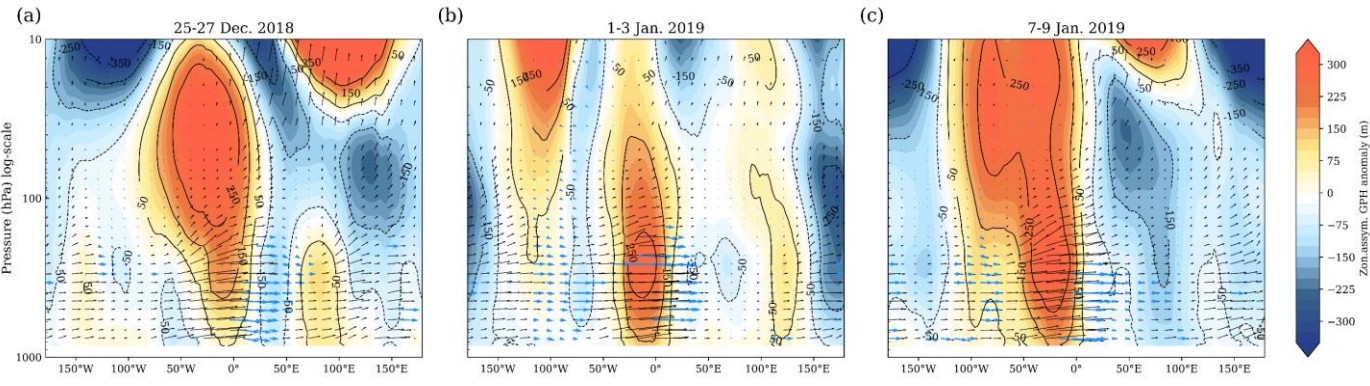

**Figure B2:** Evolution of the vertical and zonal components of the 3D Plumb flux (arrows), shown for the SSW 2019: during the SPV displacement (25-27 Dec. 2018.), during its split (1-3 Jan.) and after the split (7-9 Jan.) The vectors are plotted for every 5th longitude and pressure level. Blue vectors denote where the vertical component of Plumb flux is negative. Shading indicates the anomaly of the zonally asymmetric component of the geopotential height.