# Peer review of "Observational perspective on SSWs and blocking from EP fluxes"

_EGUsphere, 2023_

## Author Comment (AC1)

**Response to Referee #1**

(Referee comments: https://doi.org/10.5194/egusphere-2023-2916-RC1)

Manuscript: Yessimbet, K., Steiner, A. K., Ladstädter, F., and Ossó, A.: Observational perspective on SSWs and blocking from EP fluxes, EGUsphere [preprint], https://doi.org/10.5194/egusphere-2023-2916, 2024.

**The structure and content of the referee's comments are duplicated below. The authors' responses are in bold. Line numbers used in our responses refer to the original ACP Discussions paper. Text updates in the revised manuscript are in grey.**

In this paper, the authors examined eight major boreal Sudden Stratospheric Warming (SSW) events between 2007 and 2019 to understand the vertical coupling between the troposphere and stratosphere, as well as the relationship between SSWs and blocking events using Global Navigation Satellite System (GNSS) radio occultation (RO) observations. They classified the eight SSW events into two types of groups, i.e., "reflecting" events and "absorbing" events; two events fell into the former group, while the other six events fell into the latter one. The reflecting events were found to be displacement-type SSWs with a downward propagation of wave activity from the stratosphere to the troposphere during vortex recovery, accompanying the formation of blocking in the North Pacific region. On the other hand, the absorbing events were found to be split-type or mixed-type ones, showing the subsequent formation of blocking in the Euro-Atlantic region. The authors also showed an enhancement of the polar tropopause inversion layer as the result of SSWs, which was stronger for the absorbing events. These results are consistent with former studies and can actually reinforce the former results. For this point, the authors describe that "these results could help clarify the open question of whether split or displacement events trigger consistently different reactions in the tropospheric circulation, on which there is not yet consensus in the scientific community" (L366-367). If so, the authors should try to make dynamical discussions to answer this question as far as possible. Hence, I recommend the paper be revised with attention to the following details.

**We thank the reviewer for reviewing our manuscript, and for all the valuable comments and suggestions on how to improve it.**

**We included some more discussion on L366-367:**

**Nevertheless, we show that SSW reflecting/absorbing events differ in the magnitude of the downward impact (manifested e.g., in TIL variability, downward propagation of easterly wind and temperature anomalies) and correspond to specific divergent tropospheric responses. Reflecting events connected to vortex displacement are observed to trigger downward wave propagation inducing blocking over the North Pacific region while absorbing events connected to vortex split are associated with blocking over the North Atlantic and upward wave propagation. The magnitude of the downward impact may be one of the factors to consider in addressing the open question of whether displacement or split events trigger different responses in the tropospheric circulation.**

Specific Comments:

(1) L.154-156: In this analysis, eddy meridional heat fluxes are estimated at 100 hPa, while zonal mean zonal winds are calculated at 10 hPa. What does the two-day lag mean in this case? Please add comments on this point.

**The two-day lag between two consecutive heat flux pulses may indicate that the propagation of the upward wave activity (initial pulse) was somehow suppressed and then continued again, which looked like there were two two-day lagged heat flux pulses. This is part of a larger research question, namely, what enhances or suppresses the propagation of upward wave activity around 100 hPa. As for the mean zonal wind, which generally weakened due to these two pulses, it strengthened shortly during the two days between the heat flux pulses. In this regard, we added the following text in the manuscript in Line 156:**

**The two-day lag between these two pulses may be an indication that the upward wave activity was suppressed and then resumed.**

(2) L.159, 164: "Fp" should be "F_p"

**Thank you for noticing it. We corrected it in the text.**

(3) L.189-191: Negative eddy meridional heat fluxes mean the occurrence of downward propagation of wave activity. As discussed in Kodera et al. (2016), downward propagation of wave activity could enhance meandering of zonal flows, which gives a favorable condition for blocking occurrence. Therefore, the development of the North Pacific blocking might be seen during the negative eddy meridional heat flux peak. In order to more clearly see the relationship between the downward wave propagation and the blocking occurrence, I recommend the authors to make 3-d analyses by the use of 3-d wave activity fluxes, such as Plumb's (1985) one. As in Kodera et al. (2013), it might be shown that downward propagation of wave packets over North America induces a ridge over the North Pacific as well as a trough over eastern Canada in the upper troposphere.

**Thank you for the suggestion. We computed the 3D Plumb flux, according to Plumb (1985) (eq. 5.7) and we added a figure in the Appendix B (Fig.B1) showing its evolution during the SSW of 2008 (see Fig.R1.1 below). On 12-14 January, upward wave activity is enhanced over Eurasia with a clear westward tilt of the geopotential height anomalies. Starting from 21-23 February 2008, and particularly on 27-29 February we observe downward propagation of wave activity between around 250°E and 300°E (longitudinal range of North America) along with a trough centered over 300°E (longitudinal position of eastern Canada) and a positive barotropic geopotential height anomaly between 150°E and 200°E (North Pacific). This is observed together with the eastward tilt of the trough which is associated with Rossby wave downward propagation. This picture is also in agreement with Kodera et al. (2008). This, in**

turn, induced the formation of the North Pacific ridge, which then led to the formation of the North Pacific blocking.

We added more details in the manuscript at Line 205 (on new paragraph):

In addition to the EP flux analysis, to further examine the evidence for the relationship between the downward propagation of wave activity and the North Pacific blocking, we analyzed the evolution of the 3D Plumb flux (Fig. B1). Starting from 21-23 February (Fig. B1b), and particularly on 27-29 February 2008 (Fig. B1c), a downward propagation of wave activity is observed between about 250°E and 300°E along with a trough centred over 300°E and a positive barotropic geopotential height anomaly between 150°E and 200°E (North Pacific). This is observed together with the eastward tilt of the trough, implying downward propagation of the Rossby waves, aligning with the findings of Kodera et al. (2008). This in turn induced the formation of the North Pacific ridge, which then led to the formation of the North Pacific blocking.

And the new reference in the text is:

Plumb, R. A.: On the three-dimensional propagation of stationary waves. J. Atmos. Sci., 42, 217−229, https://doi.org/10.1175/1520-0469(1985)042<0217:OTTDPO>2.0.CO;2, 1985.

[Figure]

**Figure R1.1. Evolution of the vertical and zonal components of the 3D Plumb flux (arrows), shown for the SSW 2008: before the SPV displacement (12-14 Jan.), during its displacement (21-23 Feb.) and during its recovery (27-29 Feb.) The vectors are plotted for every 5th longitude and pressure level. Blue vectors denote where the vertical component of Plumb flux is negative. Shading indicates the zonally asymmetric component of the geopotential height anomaly.**

(4) L.195-204: For the reflecting events, easterlies remain only in the upper stratosphere and westerlies still dominates the lower to middle stratosphere with a maximum in the lower stratosphere, as seen Fig. 3. In such a situation, refractive index squared might be negative in the upper flank of the westerly maximum. If so, it could be a favorable condition to wave reflection

for upward propagating wave packets from the lower atmosphere, as discussed by Perlwitz and Harnik (2003). Could the authors add any discussions on this point?
**We added the following discussion in the manuscript at Line 199:**

**From 21 to 23 February, it can be observed that easterly winds are already present in the upper stratosphere, while westerly winds prevail in the middle and lower stratosphere. According to Perlwitz and Harnik (2003) and Kodera et al. (2008), this negative wind shear indicates favourable conditions for the reflection of upward propagating wave packets. When these wave packets encounter a transition from lower regions where westerly winds support their upward propagation to easterly winds that oppose it, an effective barrier to upward propagation is formed and results in the reflection of part of the wave energy.**

**We included the new reference in the reference section:**

**Perlwitz, J., and Harnik, N.: Observational evidence of a stratospheric influence on the troposphere by planetary wave reflection. J. Climate, 16, 3011–3026, https://doi.org/10.1175/1520-0442(2003)016<3011:OEOASI>2.0.CO;2, 2003.**

(5) L.244-263: As for the absorbing events, downward descending easterlies attain to the lower stratosphere, as shown in Fig. 5, which brings about a negative signal of the Northern Annular Mode in the troposphere and generally gives a favorable condition for blocking occurrence. In this case, upward wave packets, which could be occasionally originated from blocking highs, would supply easterly momentum to maintain warming events. Such a positive feedback mechanism seems to occur in these events. I also recommend the authors to make 3-d analyses here in order to show the details, as in Comment (3). In this case, it is interesting whether to see features that continuous upward wave packets originated from blockings might contribute to the continuation SSWs.

**Thank you for the suggestion. As in comment 3, we included Fig. B2 in the Supplementary Information and here below (as Fig. R1.2), showing the Plumb flux evolution for the main dynamic phases of one of the absorbing SSW events (2019 event). Indeed, it can be observed that wave activity is enhanced and directed outward from the North Atlantic blocking which is barotropically related to the positive geopotential height anomaly in the stratosphere. This suggests that the North Atlantic block may have emitted some of the wave packets into the stratosphere, thereby contributing to vortex weakening and further SSW development.**
**In this regard, we added the following text in the manuscript in Line 268:**

**To further investigate the relationship between the North Atlantic blocking and the details of wave activity propagation, the 3D Plumb flux evolution and vertically resolved geopotential height anomalies are shown in Fig. B2. Along with the onset of North Atlantic blocking formation, it can be observed that the wave activity is enhanced outward from the positive geopotential height anomalies centred between 50°W and 0°. This suggests that wave**

**packets originating from the North Atlantic block propagate into the stratosphere, thereby contributing to vortex weakening and further SSW development.**

[Figure]

**Figure R1.2. Evolution of the vertical and zonal components of the 3D Plumb flux (arrows), shown for the SSW 2019: during the SPV displacement (25-27 Dec2018.), during its split (1-3 Jan.) and after the split (7-9 Jan.) The vectors are plotted for every 5th longitude and pressure level. Blue vectors denote where the vertical component of Plumb flux is negative. Shading indicates the anomaly of the zonally asymmetric component of the geopotential height.**

(6) L.325-329: Since influences induced by the absorbing events can reach the UTLS region and make large temperature anomalies there, the polar tropopause inversion layer would be more evidently enhanced. Please add further discussions on this point.

**We added the following discussion in the text at Line 328:**

**This shows that the absorbing SSW events 2009-2019 had stronger and more prolonged impact (in terms of thermal heating) on the UTLS and the enhancement of the polar TIL than the reflecting events in 2007 and 2008.**

**References:**

**Kodera, K., Mukougawa, H., and Itoh S.: Tropospheric impact of reflected planetary waves from the stratosphere, Geophys. Res. Lett., 35, L16806, https://doi.org/10.1029/2008GL034575, 2008.**

**Perlwitz, J., and Harnik, N.: Observational evidence of a stratospheric influence on the troposphere by planetary wave reflection. J. Climate, 16, 3011–3026, https://doi.org/10.1175/1520-0442(2003)016<3011:OEOASI>2.0.CO;2, 2003.**

**Plumb, R. A.: On the three-dimensional propagation of stationary waves. J. Atmos. Sci., 42, 217−229, https://doi.org/10.1175/1520-0469(1985)042<0217:OTTDPO>2.0.CO;2, 1985.**

---

## Author Comment (AC2)

**Response to Referee #2**

(Referee comments: https://doi.org/10.5194/egusphere-2023-2916-RC2)

Manuscript: Yessimbet, K., Steiner, A. K., Ladstädter, F., and Ossó, A.: Observational perspective on SSWs and blocking from EP fluxes, EGUsphere [preprint], https://doi.org/10.5194/egusphere-2023-2916, 2024.

**The structure and content of the referee's comments are duplicated below. The authors' responses are in bold. Line numbers used in our responses refer to the original ACP Discussions paper. Text updates in the revised manuscript are in grey.**

Summary:

This work examines the structure, origin, and tropospheric influence of eight major SSW events between 2007 and 2019. All analysis is based on GNSS RO derived temperature soundings. The data are horizontally gridded to provide the geostrophic winds needed to derive factors that characterized SSW events such as EP flux and EP flux divergence. Displacement (type 1), splitting, and mixed SSW events (type 2) are compared and contrasted. The study focuses on the relation between the lower stratosphere (100 - 10 hPa), the troposphere, and tropospheric blocking during the evolution of SSW events. Results showed that GNSS RO observations are capable of capturing the main features of the SSW events. Examination of wave reflecting and absorbing SSW events highlighted the relatively short duration of the wind reversal as well as the formation of the North Pacific blocking pattern during wave reflecting events. Results also showed that the behavior of the enhanced upward EP flux prior to the SSW events differed between the two types of warming events. Furthermore, the TIL was found to depend on the magnitude of the SSW near the tropopause.

**We thank the reviewer for reviewing our manuscript. We also thank the reviewer for all the valuable comments and suggestions and for emphasizing the strengths and weaknesses of the work.**

Strengths:

The paper relates a two-dimensional, zonally averaged, dynamical analysis of SSW events to the evolution of tropospheric blocking events, providing an analysis of connections between changes in both the stratosphere and the troposphere. The detailed examination of the two main patterns identified provided a useful and engaging approach to characterizing all the warming events in the study. The dependence of the TIL on the structure of the warming event was an intriguing result.

Weakness:

1) The use of geostrophically derived winds in the stratosphere can lead to errors in the EP flux calculation. As shown in Boville (1987) the stratospheric heat flux and momentum flux errors can be as large as 40%. For example, Fig. 1 arrows at 60N and 10 hPa are nearly vertical, differing

from the NH winter climatology shown in Butchart (2022, Fig. 4a). While the citations in the text claim reasonable errors when using the geostrophic approximation, some comparison with corresponding reanalysis results should be shown to justify reliance on the geostrophic approximation, especially for the stratospheric fields.

Butchart, N.: The stratosphere: a review of the dynamics and variability, Weather Clim. Dynam., 3, 1237–1272, https://doi.org/10.5194/wcd-3-1237-2022, 2022.

Boville, B. A.: The validity of the geostrophic approximation in the winter stratosphere and troposphere, J. Atmos. Sci.,44,pp 443-457, 1987.

**Thank you for raising this important aspect. During our analysis, we compared our RO-based geostrophic parameters with non-geostrophic parameters based on reanalyses (ERA5 and NCEP). The primary information is obtained from two zonally averaged parameters such as zonal-mean zonal wind ($\bar{u}$), and eddy meridional heat flux ($\overline{v'T'}$). Figures S2,3 (also here R2.1,2.2) provide a comparison of these parameters and show consistency between RO and ERA5 both in the stratosphere and upper troposphere, confirming the reliability of RO-based dynamics.**

**Additionally, Leroy et al (2007), who computed quasi-geostrophic EP flux from RO, demonstrated that the vertical component of EP flux has a difference of only about 5-10% with the NCEP reanalysis while the horizontal component of the momentum flux has a larger difference of about 30-40%. In our work, we mainly focus on the vertical component of the EP flux because it is the vertical component that controls the stratospheric adiabatic circulation, and thus is the main informative parameter.**

**Also, our study uses geostrophic winds based on Scherllin-Pirscher et al. (2014), who showed that all major wind features are captured, and compared to atmospheric analyses, winds are only about 2 m/s weaker except near the subtropical jet where the difference is larger (up to 10 m/s).**

**We added the following information in the revised manuscript to the data description:**
**Geostrophic wind fields can be derived from RO geopotential height fields (Scherllin-Pirscher et al., 2014; 2017). RO geostrophic wind and gradient wind fields were found to capture all main wind features in our study. Compared to atmospheric analyses, wind differences are generally small (2 m/s) except near the subtropical jet (up to 10 m/s). There, RO winds underestimate actual winds due to the geostrophic and gradient wind approximations while RO retrieval errors have negligible effects (Scherllin-Pirscher et al., 2014).**

**We added also some information about the comparison of parameters between RO and ERA5 in the manuscript at Line 123:**
**In our analysis, we also made a comparison of key parameters between RO and reanalyses (e.g., ERA5), such as the zonally averaged parameters, zonal-mean zonal wind and $\overline{v'T'}$**

**(Figures S2 and S3), confirming the consistency and the reliability of the RO-based dynamics.**

[Figure]

**Figure R2.1. Zonal-mean zonal wind computed from RO (geostrophic wind; left) and ERA5 (real wind; right) and their difference for an exemplary day.**

[Figure]

**Figure R2.2. Eddy meridional heat flux computed from RO (using geostrophic meridional wind; left) and ERA5 (using real meridional wind; right) and their difference for an exemplary day.**

**Concerning the EP flux vectors, which in our manuscript (e.g., Fig.1) appear almost vertical in the stratosphere, we re-checked the calculation of the EP flux and found that the choice of the scaling of the EP flux vectors was the problem. In the initial manuscript version, we scaled the EP flux vector by the $\sqrt{1000/pressure}$, while Butchart (2022) did not use this scale factor. Once we remove this additional scale factor, the vectors look more similar to Fig.4 in Butchart (2022) (see Figure below). Regarding this scale factor, Jucker et al (2021) state that this commonly used scale factor should be used with caution and may not always be useful as it does not have any physical meaning. We decided to remove this scale factor and re-plotted all figures.**

[Figure]

**Figure R2.3.: Meridional cross section of EP flux vectors, Fig. 2c initial manuscript (left) and revised manuscript (right).**

2) It is not clear how the high vertical resolution of the GNSS RO observations contributed to the results as the vertical grid used seemed similar to current model and reanalyses. Is this work mainly a feasibility study for future work based on GNSS RO observations? More explanation can be done here.

**Sorry for having caused confusion. The main objective of our study was to get a better understanding on the vertical coupling between the troposphere and stratosphere and the relationship between SSWs and blocking events by investigating the vertical structure of planetary wave propagation, static stability, geometry of the polar vortex, and the occurrence of blocking events from an observational viewpoint. We therefore position this study as an extension of observational knowledge of stratosphere-troposphere dynamics and SSW events.**

The feasibility of GNSS RO has already been investigated by several studies as stated and cited in the introduction section of the manuscript (see the references below). However, we have also made further comparisons with reanalyses, confirming the consistency of the RO-based dynamics (see response to comment 1).

In addition, we compared the Brunt Väisälä frequency (Fig. S4/R3.4) which shows high consistency in terms of main patterns and magnitude between RO and ERA5. Small differences (of around 10%) in $N^2$ are observed mainly in the tropopause region between 200 hPa and 300 hPa and in the stratosphere.

It should be noted that we chose to compare with ERA5 because it is arguably the most advanced and commonly used reanalysis product, however, ERA5 assimilates RO data. RO data has a high vertical resolution, while the daily gridded field is smoothed in the horizontal and over time due to weighted averaging. Therefore, it is not straightforward to interpret the differences in detail.

However, for our study we decided to take the observational perspective and chose to use GNSS RO observations for the analyses as the dataset resolves the relevant features to provide information on the stratosphere–troposphere coupling.

For better clarity we revised the manuscript text, specifically the first and last paragraph in the discussion section, which now read:
The main objective of this study was to characterize the synoptic and dynamic conditions of SSWs and to investigate the link to blocking events from an observational perspective. We used GNSS RO observation for these analyses as the dataset resolves the relevant features to provide information on the stratosphere–troposphere coupling.

In conclusion, our findings underscore the applicability of GNSS RO for the exploration of atmospheric circulation dynamics. Due to its high vertical resolution, GNSS RO has the potential for studying the interplay between tropopause structure and wave activity propagation.

References:

Leroy, S. S., and J. G. Anderson (2007), Estimating Eliassen–Palm flux using COSMIC radio occultation, Geophys. Res. Lett., 34, L10810, doi:10.1029/2006GL028263

Scherllin-Pirscher, B., A. K. Steiner, and G. Kirchengast (2014) Deriving dynamics from GPS radio occultation: Three-dimensional wind fields for monitoring the climate. Geophys. Res. Lett., 41, 7367–7374, doi:10.1002/2014GL061524.

Scherllin-Pirscher, B., A. K. Steiner, G. Kirchengast, M. Schwärz, and S. S. Leroy (2017), The power of vertical geolocation of atmospheric profiles from GNSS radio occultation, J. Geophys. Res. Atmos., 122, 1595–1616, doi:10.1002/2016JD025902.

**Verkhoglyadova, O., S. Leroy, and C. Ao (2014), Estimation of winds from GPS radio occultations, J. Atmos. Oceanic Technol., doi:10.1175/JTECH-D-14-00061.1.**

Minor Comment:

Line 151: Kordera et al., 2016 define recovery at 50 hPa, however, the vertical line denoting recover appears to based on 10 hPa temperatures in Fig. 3a. Some explanation is needed.

**The vertical line in Fig.3a shows exactly when the temperature at 50 hPa is at its maximum.**

Recommendation: Publish after address the two weaknesses noted above.

**Thank you.**

**References:**

**Leroy, S. S., and Anderson, J. G.: Estimating Eliassen–Palm flux using COSMIC radio occultation, Geophys. Res. Lett., 34, L10810, doi:10.1029/2006GL028263, 2007.**

**Scherllin-Pirscher, B., Steiner, A. K., and Kirchengast, G.: Deriving dynamics from GPS radio occultation: Three-dimensional wind fields for monitoring the climate, Geophys. Res. Lett., 41, 7367–7374, https://doi.org/10.1002/2014GL061524, 2014.**

**Scherllin-Pirscher, B., A. K. Steiner, G. Kirchengast, M. Schwärz, and S. S. Leroy (2017), The power of vertical geolocation of atmospheric profiles from GNSS radio occultation, J. Geophys. Res. Atmos., 122, 1595–1616, doi:10.1002/2016JD025902.**

**Jucker, M.: Scaling of Eliassen-Palm flux vectors, Atmos. Sci. Lett., 22, e1020, https://doi.org/10.1002/asl.1020, 2021.**

**\*Figure S4 from Supplementary Information**

[Figure]

**Figure S4. Brunt Väisälä frequency computed from RO (a) and ERA5 (b) and their difference (c) and difference in percentage (d) averaged over 75-90° N within a +/- 30 day**

**timeframe relative to each of the SSW events from 2007 to 2019. Hatched regions indicate dates when the zonal-mean zonal wind at 60° N and 10 hPa is negative.**

---

## Author Comment (AC4)

**Response to Referee #3**

(Referee comments: https://doi.org/10.5194/egusphere-2023-2916-RC3)

Manuscript: Yessimbet, K., Steiner, A. K., Ladstädter, F., and Ossó, A.: Observational perspective on SSWs and blocking from EP fluxes, EGUsphere [preprint], https://doi.org/10.5194/egusphere-2023-2916, 2024.

**The structure and content of the referee's comments are duplicated below in regular font. The authors' responses are in bold. Line numbers used in our responses refer to the original ACP Discussions paper. Text updates in the revised manuscript are in grey.**

This paper examines the use of GNSS radio occultation (RO) data for investigating key features of sudden stratospheric warmings (SSWs). The authors grid RO data following established techniques and then calculate geostrophic winds from this gridded product. Wave diagnostics – meridional heat flux and Eliassen-Palm (EP) flux – are then extracted from these data. SSWs are identified and classified based off these data, and the evolution of wave forcing and vertical coupling are shown. The authors provide thorough discussion about this evolution for sets of known SSW types: displacement and split events, and reflecting and absorbing events. They acknowledge their sample size is small, but find evidence in support of previous work that displacement events are followed by North Pacific blocking and split events are followed by North Atlantic blocking.

The work is presented well, though the authors could consider some rearrangement of two figures. This work strictly uses observational data – along with dynamical theory – which is still somewhat novel. Other observational studies have looked at more limited samples or other features of SSWs such as only the stratospheric fields. And the thorough, deep analysis of a small, but representative set of events provides useful information for the community. The authors adeptly touched on several outstanding issues in our understanding of the wave-mean flow interaction and stratosphere-troposphere coupling that occurs around SSWs.

However, the manuscript would benefit greatly from additional work on a few topics. These are mostly related to how the GNSS data compare with other, well-studied data sets and what the GNSS data provide that is new.

**We thank the reviewer for reviewing our manuscript and for providing constructive comments and advice on how to improve it. We also thank the reviewer for the positive comments on our manuscript, which emphasized the value and novel aspects of our work.**

1) Details of the GNSS RO data are missing. As the authors surely know, the RO method observes bending of signals through the atmosphere, which is foremost related to changes in density. This knowledge can then be used to derive geopotential height and pressure to a high degree of accuracy through the hydrostatic equation. Following this, temperature may be calculated, but only by assuming no moisture – i.e., "dry temperature." This is a reasonable assumption in the stratosphere but will lead to large inaccuracies in the tropospheric fields. Given how much of the discussion and results rely on temperature below 300 hPa, the authors should address the limitations of its

use. This is covered in the references they cite but the reader of this work would benefit from additional, relevant information here.

Alternatively, accurate temperature and humidity fields may be derived from RO using one-dimensional variational analysis. But it's not clear if the authors used such a product, in which case the limitations of that needs to be addressed.

**Thank you for pointing to this. We included more information on the GNSS RO data retrieval, variables, and characteristics. In our study, we use physical temperature based on a moist-air retrieval, and not the "dry" temperature.**

**In this regard, the following text is added in the Data Section at Line 76:**

**The GNSS RO method is based on the detection of radio signals transmitted by GNSS satellites, which are refracted by the Earth's atmosphere as they propagate through it to Low Earth Orbit (LEO) satellites. The measured signal phase changes are converted to bending angle profiles, and further to refractivity by an Abel transform. At high altitudes, the Abel integral requires initialization with background data. Thermodynamic parameters are then computed under the assumption of a dry atmosphere ("dry" parameters). In moist air conditions (lower to middle troposphere, specifically in the tropics), the retrieval of (physical) temperature or humidity requires prior knowledge of the state of the atmosphere (e.g., Kursinski et al. 1995; 1996). Due to the involved background data, the retrieved RO temperature data exhibit larger uncertainties in lowermost moist parts of the troposphere and at high altitudes (above about 30 km). For an overview of the retrieval process and the involved structural uncertainties see, e.g., Steiner et al. (2020). The RO measurements are of very high quality with minimal structural uncertainty within the UTLS region, as highlighted by Scherllin-Pirscher et al. (2017) and Steiner et al. (2020).**

**In this work, we use geopotential height and physical temperature as a function of pressure, processed by the Wegener Center for Climate and Global Change (WEGC) with the Occultation Processing System (OPS) version 5.6 (Angerer et al., 2017; Steiner et al., 2020).**

**Geostrophic wind fields can be derived from RO geopotential height fields (Scherllin-Pirscher et al., 2014; 2017). RO geostrophic wind and gradient wind fields were found to capture all main wind features in our study. Compared to atmospheric analyses winds, differences are in general small (2 m/s) except near the subtropical jet (up to 10 m/s). There, RO winds underestimate actual winds due to the geostrophic and gradient wind approximations while RO retrieval errors have negligible effects (Scherllin-Pirscher et al., 2014).**

**We included the following new references in the revised manuscript:**
**Kursinski, E. R., and G. A. Hajj, K. R. Hardy, L. J. Romans, and J. T. Schofield, 1995: Observing tropospheric water vapor by radio occultation using the Global Positioning System. Geophys. Res. Lett., 22, 2365–2368, https://doi.org/10.1029/95GL02127.**

Kursinski, E. R., Hajj, G. A., Bertiger, W. I., Leroy, S. S., Meehan, T. K., Romans, L. J., Schofield, J. T., McCleese, D. J., Melbourne, W. G., Thornton, C. L., Yunck, T. P., Eyre, J. R., and Nagatani, R. N.: Initial Results of Radio Occultation Observations of Earth's Atmosphere Using the Global Positioning System, Science, 271, 1107–1110, 1996.

At Line 76, we also edited the sentence "We use temperature and geopotential height profiles…":

In this work, we use geopotential height and physical temperature as a function of pressure, processed by the Wegener Center for Climate and Global Change (WEGC) with the Occultation Processing System (OPS) version 5.6 (Angerer et al., 2017; Steiner et al., 2020).

2) RO profile density may be an important topic to document for this study, but no details are given. RO missions and profile counts vary with time, but some measure of the sampling density should be given. Given the nature of RO sampling, this can likely be well-represented by zonal mean statistics. It would also be useful for the authors to document the occurrence, or likely rarity, of grid points that are missing RO profiles.

Regarding the number of RO profiles used in our study, we added Figure S1 (in Supplementary Information)/R3.1 showing the zonally averaged monthly distribution of RO profiles used in our studies. We have also added more detailed information in the manuscript in Line 82:

The number of daily RO profiles from different missions varied over the period from 2006 to 2019, with the highest number of profiles from 2007 to 2010 ( > 2500 profiles per day) and a decrease in the number of profiles (from more than 2500 to less than 2000 profiles) from 2012 onwards (Figure S1) due to the exceeding of the lifetime of some of the RO missions (Fig. 5, Angerer et al., 2017).

Thus, in the range of vertical pressure levels from 10 to 850 hPa, there are fewer than 10 missing grid points in the daily gridded fields, with the number increasing towards the surface.

[Figure]

**Figure R3.1. Zonal distribution of the monthly number of the RO profiles averaged over 10 to 850 hPa for the period from September 2006 December 2019.**

3) Limited comparisons with reanalyses may be a real benefit to the manuscript. As it stands, the manuscript doesn't give the reader a sense of what the relatively high vertical resolution of the RO observations adds to our understanding of SSWs. This is most evident in the tropopause inversion layer (TIL) results. Details of the Brunt Vaisala frequency N^2 would seem to be most sensitive to vertical resolution, and RO may be able to provide additional insight, but it's not clear what that is.

Some comparison of N^2 with, say, ERA5 for all or a limited sample of SSWs may support the authors' claims on the benefits of the high vertical resolution of RO observations.

Additionally, the authors might consider adding one or two sentences about how their diagnosed dates of the SSWs compare with other, reanalysis-based studies.

**Thank you for this suggestion. During our analysis, we compared our RO-based parameters with parameters based on reanalyses (ERA5 and NCEP), e.g., zonally averaged parameters such as zonal-mean zonal wind ( $\overline{u}$ ), and eddy meridional heat flux ( $\overline{v'T'}$ ). Figures S2,3 (also here R3.2,3.3) provide a comparison of these parameters and show consistency between RO and ERA5 both in the stratosphere and upper troposphere, confirming the reliability of RO-based dynamics.**

**As you suggest, we also include Figure S4/R3.4 depicting the Brunt Väisälä frequency computed from RO and ERA5 (averaged over 75-90°N), which shows high consistency in terms of main patterns and magnitude between RO and ERA5. Small differences (of around 10%) in $N^2$ are observed mainly in the tropopause region between 200 hPa and 300 hPa and in the stratosphere. It should be noted that we chose to compare with ERA5 because it is arguably the most advanced and commonly used reanalysis product, however, ERA5 assimilates RO data. RO data has a high vertical resolution, while the daily gridded field is**

smoothed in the horizontal and over time due to weighted averaging. Therefore, it is not straightforward to interpret the differences in detail.

However, for our study we decided to take an observational perspective and chose to use GNSS RO observations for the analyses as the dataset resolves the relevant features to provide information on the stratosphere–troposphere coupling.

We added information about the comparison of parameters between RO and ERA5 in the manuscript on Line 123:
In our analysis, we also made comparisons of key parameters between RO and reanalyses (e.g., ERA5), such as the zonally averaged parameters, zonal-mean zonal wind and $\overline{v'T'}$ (Figure S2 and S3), confirming the consistency and the reliability of the RO-based dynamics.

For better clarity with respect to the observations we revised the first and last paragraph of the discussion section, it reads now:
The main objective of this study was to characterize the synoptic and dynamic conditions of SSWs and to investigate the link to blocking events from an observational perspective. We used GNSS RO observation for these analyses as the dataset resolves the relevant features to provide information on the stratosphere–troposphere coupling.

In conclusion, our findings underscore the applicability of GNSS RO for the exploration of atmospheric circulation dynamics. Due to its high vertical resolution, GNSS RO has the potential for studying the interplay between tropopause structure and wave activity propagation. A detailed study of the relationship between tropopause structure and wave activity propagation relevant to SSW events should be investigated in future GNSS RO studies.

[Figure]

[Figure]

[Figure]

**Figure R3.2. Zonal-mean zonal wind computed from RO (geostrophic wind; left) and ERA5 (real wind; right) and their difference for an exemplary day.**

[Figure]

**Figure R3.3. Eddy meridional heat flux computed from RO (using geostrophic meridional wind; left) and ERA5 (using real meridional wind; right) and their difference for an exemplary day.**

[Figure]

**Figure R3.4. Brunt Väisälä frequency computed from RO (upper plot) and ERA5 (middle plot) and their difference (second lower plot) and difference in percentage (lowest plot) averaged over 75-90° N within a +/- 30 day timeframe relative to each of the SSW events**

**from 2007 to 2019. Hatched regions indicate dates when the zonal-mean zonal wind at 60° N and 10 hPa is negative.**

**Regarding the diagnosed dates of SSWs, we added the following sentence at Line 130:**
**The diagnosed central SSWs are compared with the list of major midwinter SSWs in the reanalysis products of the SSW Compendium dataset (NOAA CSL, 2024).**
**In the reference section we cite the SSW Compendium:**
**NOAA CSL: Chemistry & Climate Processes: SSWC,**
**https://csl.noaa.gov/groups/csl8/sswcompendium/majorevents.html**

4) Line 96: The citation to Scherllin-Pirscher et al. (2014) is not included in the references section.
**The citation to Scherllin-Pirscher et al. (2014) is already included in the references section.**

5) Figures 6 and 7 could benefit from vertical stacking into two rows of 4 panels. As they're presented, some of the details are squished into a narrow space.
**Fig. 6 is a final combination of 8 detailed figures already shown for each SSW event in the study (e.g. Fig. 3d, Fig. 5d, Fig. A2d, etc.). We therefore decided to leave Fig. 6 as it is. In Fig. 7 the main patterns seem to be clear. Also, for the sake of consistency with Fig. 6, we decided to keep Fig. 7 as it is.**

6) Line 114: Suggest starting a new paragraph at "A standard algorithm…"

**We started a new paragraph at "A standard algorithm…".**

7) Line 150: This final sentence of this paragraph feels more appropriate in the previous section with other definitions.

**We moved this sentence to the Method Section on Line 130.**

8) Line 155: Recommend "concurrent with" rather than "due to" as the heat flux is a proxy for the wave activity flux that drives the zonal wind reversal.

**Thank you. We changed the words "due to" to "concurrent with".**

You may consider a similar slight wording change on line 199: "led to a deceleration."

**We changed the wording from "led to deceleration" to "resulted in a slowing down".**

---

## Author Response (AR2)

**Author's Response**

Manuscript: Yessimbet, K., Steiner, A. K., Ladstädter, F., and Ossó, A.: Observational perspective on SSWs and blocking from EP fluxes, EGUsphere [preprint], https://doi.org/10.5194/egusphere-2023-2916, 2024.

**The structure and content of the referee's and the Editor's comments are duplicated below. The authors' responses are in bold. Line numbers used in our responses refer to the revised ACP Discussions paper. Text updates in the revised manuscript are in grey.**

**We thank the referees for their review of our manuscript. We also thank the Editor for the handling of our manuscript. We have addressed the technical corrections to the manuscript according to the comments of referees 2 and 3.**

**Comments by the editor**

All 3 referees report that their comments on the first version of the paper have been effectively addressed in revision. I am therefore pleased to accept it for publication in ACP. Two of the referees each recommend one further small change -- those can be addressed as 'technical corrections' prior to publication.

**Comments of the referee 1:**

Comments on "Observational perspective on SSWs and blocking from EP fluxes" (revised version) by Kamilya Yessimbet, Andrea K. Steiner, Florian Ladstädter, Albert C. Ossó

Recommendation: Acceptable in present form

In this paper, the authors examined eight major boreal Sudden Stratospheric Warming (SSW) events between 2007 and 2019 to understand the vertical coupling between the troposphere and stratosphere, as well as the relationship between SSWs and blocking events using Global Navigation Satellite System (GNSS) radio occultation (RO) observations. They classified the eight SSW events into two types of groups, i.e., "reflecting" events and "absorbing" events; two events fell into the former group, while the other six events fell into the latter one. The reflecting events were found to be displacement-type SSWs with a downward propagation of wave activity from the stratosphere to the troposphere during vortex recovery, accompanying the formation of blocking in the North Pacific region. On the other hand, the absorbing events were found to be split-type or mixed-type ones, showing the subsequent formation of blocking in the Euro-

Atlantic region. The authors also showed an enhancement of the polar tropopause inversion layer as the result of SSWs, which was stronger for the absorbing events. These results are consistent with former studies and can actually reinforce the former results.

The presented results are considered to include new findings and the authors revised the original description and added relevant analyses and discussions, following the reviewers' comments. Hence, I consider that the manuscript should be accepted for publication in ACP.

**Thank you.**

**Comments of the referee 2:**

On a minor note: the vertical line in Fig. 3a still appears to this reviewer to not line up with the maximum temperature at 50 hPa as stated in the response. Perhaps a horizontal line at 50 hPa would help.

**In Fig.3a, the time-height evolution of the temperature anomaly is shown for the zonal average of the temperature over 65-90° N, while the vertical line indicates the day when the polar (80-90° N) temperature anomaly reaches its maximum, i.e. the start of the SSW recovery phase. This is also stated in the figure caption. Therefore, we decided to leave the figure as it is.**

**Comments of the referee 3:**

The authors have addressed my comments and I find the manuscript to be much improved. I recommend publication.

However, I suggest the authors check line 353: the anomalies of N^2 aren't shown for each SSW, but aren't the absolute values now shown in the supplement?

**Thank you for noticing it. Yes, the absolute values of $N^2$ are now given in the supplement. So, we have corrected this in the manuscript on line 353, and the sentence now reads as follows:**

**We also note the descending enhancement of static stability from the stratosphere to the tropopause level during the onset of the SSWs, which is observed in the static stability anomalies for the 2009, 2016, 2018, and 2019 SSWs and in its absolute values for all SSWs (Fig.S4a).**